# LEARNING TO HELP IN MULTI-CLASS SETTINGS

**Yu Wu§, Yansong Li‡=, Zeyu Dong†=, Nitya Sathyavageeswaran§, Anand D. Sarwate§***
§Rutgers University, ‡University of Illinois Chicago, †Stony Brook University

## ABSTRACT

Deploying complex machine learning models on resource-constrained devices is challenging due to limited computational power, memory, and model retrainability. To address these limitations, a hybrid system can be established by augmenting the local model with a server-side model, where samples are selectively deferred by a rejector and then sent to the server for processing. The hybrid system enables efficient use of computational resources while minimizing the overhead associated with server usage. The recently proposed Learning to Help (L2H) model proposed training a server model given a fixed local (client) model. This differs from the Learning to Defer (L2D) framework which trains the client for a fixed (expert) server. In both L2D and L2H, the training includes learning a rejector at the client to determine when to query the server. In this work, we extend the L2H model from binary to multi-class classification problems and demonstrate its applicability in a number of different scenarios of practical interest in which access to the server may be limited by cost, availability, or policy. We derive a stage-switching surrogate loss function that is differentiable, convex, and consistent with the Bayes rule corresponding to the 0-1 loss for the L2H model. Experiments show that our proposed methods offer an efficient and practical solution for multi-class classification in resource-constrained environments. 🎧

## 1  INTRODUCTION

Machine Learning (ML) models deployed on local devices often face significant limitations in terms of computational resources and retrainability. Local devices are typically constrained by limited processing power, memory, and battery life (Ajani et al., 2021; Biglari & Tang, 2023), which can impede the model's ability to handle large-scale data or perform complex computations in real-time. Furthermore, once a local model is deployed, it may be difficult to retrain or update (Hanzlik et al., 2021), leading to a potential degradation in performance over time when data distribution drifts (Lu et al., 2019).

To address these issues, one strategy is to augment the local machine learning system (a "client") with an external model hosted on a remote server. This approach, seen in recent applications like *Apple Intelligence* (Gunter et al., 2024), enhances the overall system performance by leveraging the server's superior computational power and capacity for model updates. Recent studies have shown that efficient fine-tuning methods, like few-shot learning (Brown et al., 2020) and parameter-efficient fine-tuning (Fu et al., 2023), can achieve competitive performance, making it a feasible choice for server-side model deployment. Those methods utilize the pre-trained model's capabilities while requiring minimal computational resources, allowing for easy and cost-effective updates to adapt to specific tasks or data distributions. Consequently, it enables the deployment of powerful models on servers without the high costs of comprehensive model training.

While training a server model can be relatively inexpensive, extensive use of server-side models can be costly due to data transfer, latency, instability connection, and service fees. Moreover, server operators may wish to limit the frequency of offloaded inferences from clients by imposing costs

---

*Corresponding author: anand.sarwate@rutgers.edu; =Equal contribution
This material is based upon work supported by the US National Science Foundation under grant CNS-2148104 and is supported in part by funds from federal agency and industry partners as specified in the Resilient & Intelligent NextG Systems (RINGS) program.

or access constraints. In the presence of such restrictions, the practical solution for clients to use a *rejector* that selectively defers/offloads only the most challenging and uncertain samples to the server for processing, thereby optimizing the balance between usage cost and performance. Informally, the rejector is defined as a decision function to either select the local model or server model for a sample (a rigorous definition is given in Sec. 2).

A framework for training a rejector and server model given a fixed "legacy" device-bound model was recently proposed for binary classification models under the name "Learning to Help" (L2H) (Wu & Sarwate, 2024). In this work, we extend and generalize this framework to encompass a variety of problems of more practical interest. We can identify several scenarios that can arise in the context of building ML systems in those settings with a fixed local model. In particular, we demonstrate how to handle various restrictions on the use of the server:

- PAY-PER-REQUEST (PPR): the device must pay a cost each time the rejector defers to the server; as an example, the server can be thought of as a consultant who offers their services at a cost.

- INTERMITTENT AVAILABILITY (IA): in the PPR model, the rejection rule must also account for the server being potentially unavailable; for example, this can occur when the internet connection is unstable, or the server is busy servicing requests of other devices.

- BOUNDED REJECT RATE (BRR): instead of PPR, the rate of rejections/deferrals per unit time may not exceed a predefined upper limit; for example, there is a limit on the usage of some LLMs even with a paid subscription.

The L2H model is a natural complement to prior works on two-party decision systems, including *Learning with Abstention* (LWA) and *Learning to Defer* (L2D). These models either assume that rejected samples can be discarded at a fixed cost or that the server is a pre-trained "expert" and optimize decision rules for the device/rejector. As ML/AI decision systems become integrated into physical devices and infrastructure, issues of sustainability will require newer server systems to support pre-existing legacy models which can allow only partial retraining. Relatively little attention has been given to this scenario: we address the extreme case where only the rejection rule can be updated with the server.

Existing work (Wu & Sarwate, 2024) on the L2H model only studies binary classification in the PPR model. In this work, we further extend the L2H framework to multi-class problems and show how to train systems under the three scenarios mentioned above. More specifically, we propose a generalized (non-differentiable) 0-1 loss to measure the prediction system's performance and find the Bayes rule for this loss. We then provide a surrogate loss function which is convex and differentiable and show that its minimizer is consistent with the Bayes rule for the generalized 0-1 loss.

We then design algorithms which can optimize rules for training multi-class predictors in the three scenarios mentioned earlier: PPR, IA, and BRR. To guarantee BRR we use a "post-hoc" method to control the rejection rate. We show experimentally that incorporating a server model with a rejector enhances the overall performance of the ML system in all three scenarios. Training with our surrogate loss function, the rejector helps in identifying the challenging and uncertain samples, while balancing the accuracy and usage of the server model.

RELATED WORKS

**Hybrid ML systems.** The hybrid ML systems that consist of client side and server side, have been of substantial research in many fields, including federated learning (Zhang et al., 2021; McMahan et al., 2017), distributed learning (Cao et al., 2023a; Horváth et al., 2023), and decentralized learning (Sun et al., 2022; Fang et al., 2022; Liu et al., 2024; Li & Han, 2023). However, those frameworks only focus on interaction between different sides during the training process. Once the training is done, no more communication is needed among them. In this work, we are interested in the collaboration between client and server both in the training and inference phases.

**Learning with Abstention (LWA).** The foundational work that first considered extra options for recognition tasks was proposed by Chow (1957) and Chow (1970). Herbei & Wegkamp (2006) revisit the framework in classification tasks and propose a score-based reject method. Subsequent works extend this framework to different types of classifiers (Rigollet, 2007; Wegkamp, 2007;

Bartlett & Wegkamp, 2008; Wegkamp & Yuan, 2011). Cortes et al. (2016) consider a separate function to make reject decisions in binary classification. Zhu & Nowak (2022b;a) introduce the reject option in active learning. Cortes et al. (2018) add abstention to online learning. Zhang et al. (2024) consider the rejection rule when the data distribution contains noisy labels. Yin et al. (2024) use rejection for ensuring fairness. Mao et al. (2024d) and Mao et al. (2024c) analyze the theoretical bounds and algorithms for score-based and predictor-rejector-based methods, respectively.

Unlike the setting where rejection incurs an extra cost, few works focus on scenarios with no extra cost for rejection, but the reject rate is bounded. Pietraszek (2005) construct the rejection rule using ROC analysis. Denis & Hebiri (2020) derive the Bayes optimal classifier for the bounded reject rate setting. Shekhar et al. (2020) consider binary classification with bounded reject rates in the active learning setting. In LWA, although there are works that can identify data samples that are uncertain or challenging for the local model to predict, the subsequent action is merely to discard those samples without providing an answer. While this framework can indeed improve prediction accuracy, the system still cannot handle the samples that are challenging for the local model.

**Learning to Defer (L2D).** Building upon LWA, Madras et al. (2018) were the first to consider a subsequent expert can process the rejected samples. Raghu et al. (2019) consider binary classification with expert deferral using uncertainty estimators. Mozannar & Sontag (2020a) propose the first method that jointly trains the local model and rejector in multi-class classification. Verma & Nalisnick (2022); Mozannar et al. (2023); Hemmer et al. (2023); Cao et al. (2023b) propose different surrogate loss functions for L2D. Okati et al. (2021) and Narasimhan et al. (2022) add post-hoc algorithms to the cases where the reject decision incurs extra cost.

Seeking help from multiple experts has been explored in different machine learning models (Kerrigan et al., 2021; Keswani et al., 2021; Corvelo Benz & Gomez Rodriguez, 2022; Hemmer et al., 2022). Verma et al. (2022) extend the one-vs-all-parameterized L2D to the multiple experts case. Mao et al. (2023) study a two-stage scenario for L2D with multiple experts. Mozannar et al. (2023) provide a linear-programming formulation that optimally solve L2D in the linear setting. Verma et al. (2023) incorporate a conformal inference technique for multiple experts. Tailor et al. (2024) formulate a L2D system that can cope with never-before-seen experts. Mao et al. (2024b) introduce regression with deferral to multiple experts. Mao et al. (2024a) present a theoretical study of surrogate losses and algorithms for L2D with multiple experts. L2D complements the missing part of LWA; that is, after a reject decision, instead of being discarded, samples are sent to remote experts for predictions. However, in L2D, the experts on the server side are assumed to be either human or well-trained ML models, which are fixed during the training process. L2D focuses on training the local model and the rejector under the existence of expert models. This framework cannot handle the cases described in Sec. 1, where local models are legacy systems that have been deployed to the client side, and retraining the local models is not available.

**Learning to Help (L2H).** Learning to Help (L2H) (Wu & Sarwate, 2024) is a complementary model to L2D whose aim is to train a server model and rejector to enhance the usability of systems containing a "legacy" local ML model that is unavailable for updating or retraining. The prior work focused on binary classification in the PPR setting and proposes a surrogate loss function that requires model-based calibration. This cannot be directly extended to multi-class problems due to issues with differentiability. In this work we generalize this model to multi-class problems and more scenarios (see Appendix A for a comparison).

## 2  PROBLEM FORMULATION

We consider a decision system with two parties: a *client* and the *server*. We consider the client a device with relatively limited computational power while the server has access to more computing resources. We study a multi-class problem in which the goal is to learn a prediction function $f\colon \mathcal{X} \to \mathcal{Y}$, where $\mathcal{X} \subset \mathbb{R}^l$ is a space of instances/feature vectors with $l$ dimensions and $\mathcal{Y} = [K] \triangleq \{1, 2, \ldots, K\}$ is a set of labels.

The L2H decision system (See Fig. 1) is constrained to have a certain architecture: in operation, the client receives an input $x$ and has two options: it can either make a prediction $\hat{y}_{\text{local}}$ locally on the client or forward the input ("defer") $x$ to the remote server which produces a response $\hat{y}_{\text{remote}}$.

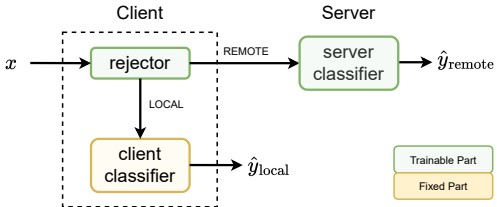

Figure 1: Diagram of learning to help framework.

We can represent this system by a tuple of functions $(r(x), m(x), e(x))$, where the client side has a reject function $r(x)$ and prediction function $m(x)$ and the server has a prediction function $e(x)$. The function $r(x)$ makes a binary choice representing the client's choice to use from those two options and then lets either $m(x)$ or $e(x)$ make a prediction for $x$.

We assume that the training algorithm has access to training set $\{(x_i, y_i) : i \in [n]\}$ of $n$ feature-label pairs in $\mathcal{X} \times \mathcal{Y}$. For the purposes of analysis we make the standard assumption (Vapnik, 2010; Shalev-Shwartz & Ben-David, 2014) that random variables of feature vector $X$ and label $Y$ are sampled independently and identically distributed (i.i.d.) according to an unknown distribution $\mathcal{D}$.

The function $r\colon \mathcal{X} \to \{\text{LOCAL}, \text{REMOTE}\}$, known as the *rejection rule* or *rejector*, determines whether the client should label the input locally or send it to the remote server for labeling. Formally, if $r(x) = \text{LOCAL}$, the client uses the local classifier $m(x)$ to provide a label. Conversely, if $r(x) = \text{REMOTE}$, the client forwards sample $x$ to the server, which then labels the input using remote classifier $e(x)$.

For the functions $r(x)$, $m(x)$, and $e(x)$, there are four possible outcomes for a given sample $(x, y)$. The first event occurs when $r(x) = \text{LOCAL}$ and $m(x) = y$, representing a correct decision by the client. The second event is when $r(x) = \text{LOCAL}$ and $m(x) \neq y$ indicate an error by the client. The third event occurs when $r(x) = \text{REMOTE}$ and $e(x) = y$, representing a correct decision by the server. The fourth event is when $r(x) = \text{REMOTE}$ and $e(x) \neq y$, indicate an error by the server. We define the costs associated with each of these outcomes as $c_{\text{cc}}$, $c_{\text{ce}}$, $c_{\text{sc}}$, and $c_{\text{se}}$, respectively. For the scenarios discussed in Sec. 1, our goal is to provide accurate predictions, satisfying $c_{\text{cc}} \leq c_{\text{ce}}$ and $c_{\text{sc}} \leq c_{\text{se}}$, as well as instant predictions, satisfying $c_{\text{cc}} \leq c_{\text{sc}}$ and $c_{\text{ce}} \leq c_{\text{se}}$. Based on four outcomes and corresponding costs, the general loss function of this two-side framework given sample $(x, y)$ is defined as:

$$L(r, e, x, y, m) = c_{\text{cc}} \mathbf{1}_{m(x)=y} \mathbf{1}_{r(x)=\text{LOCAL}} + c_{\text{ce}} \mathbf{1}_{m(x) \neq y} \mathbf{1}_{r(x)=\text{LOCAL}} \tag{1}$$
$$+ c_{\text{sc}} \mathbf{1}_{e(x)=y} \mathbf{1}_{r(x)=\text{REMOTE}} + c_{\text{se}} \mathbf{1}_{e(x) \neq y} \mathbf{1}_{r(x)=\text{REMOTE}},$$

where $\mathbf{1}_{[\cdot]}$ is the indicator function. The expected loss, also known as risk, is defined as $R(r, m, e) \triangleq \mathbf{E}_{(X,Y) \sim \mathcal{D}}[L(r, e, x, y, m)]$. The *Bayes Classifiers* then defined as: $r^B, m^B, e^B \in \arg\min_{r,m,e} R(r, m, e)$. Jointly training all three functions is trivial because the Bayes classifiers satisfy $m^B = e^B = \arg\max_i \eta_i(x)$, where

$$\eta_i(x) = P(Y = i \mid X = x) \tag{2}$$

represents the regression function for $i$-th class. The proof is similar with Proposition 2 derived by Mozannar & Sontag (2020b). Since the Bayes client classifier equals the Bayes server classifier, together with the cost constraints stated above, we have $r^B(x) = \text{LOCAL}$, for all $x$. No input will be sent to the server.

Since jointly training three functions leads to a trivial solution, we focus on a constrained but still interesting case, *learning to help* (L2H), where the client classifier cannot be updated or retrained (Fig. 1). While the prior work on L2H only considers binary classification, in this paper we extend the approach to multi-class problems.

## 2.1 GENERALIZED 0-1 LOSS FOR MULTI-CLASS CLASSIFICATION

To extend L2H to multi-class classification, we consider a special case of the general loss function (equation 1), often referred to as the generalized 0-1 loss in binary classification for L2H. In the

generalized $0$-$1$ loss, the parameters are defined as follows: $c_{\text{cc}} = 0$, $c_{\text{ce}} = 1$, $c_{\text{sc}} = c_{\text{e}}$, and $c_{\text{se}} = c_{\text{e}} + c_1$. The idea behind this setup is that there is no cost when the local prediction is correct, but a cost of 1 is incurred when the local prediction is incorrect. Therefore, we set $c_{\text{cc}} = 0$ and $c_{\text{ce}} = 1$. Additionally, recall the interesting settings we mentioned in Sec. 1; requesting assistance from a remote server is not free. Once a sample is sent to the server, a constant *reject cost* $c_{\text{e}}$ is incurred, regardless of the server's prediction. Therefore, $c_{\text{sc}} = c_{\text{e}}$. Furthermore, if the server prediction is incorrect, an additional penalty *inaccuracy cost* $c_1$ is imposed to account for the mistake. It is important to note that $c_1$ may be greater than 1, especially in critical scenarios such as medical diagnoses, where expert misdiagnoses can lead to more severe consequences. As a result, we set $c_{\text{se}} = c_{\text{e}} + c_1$. In summary, the generalized $0$-$1$ loss for multi-class classification for L2H is defined as:

$$L_{\text{general}}(r, e, x, y; m) = \mathbf{1}_{m(x) \neq y} \mathbf{1}_{r(x) = \text{LOCAL}}$$
$$+ c_{\text{e}} \mathbf{1}_{e(x) = y} \mathbf{1}_{r(x) = \text{REMOTE}} + (c_{\text{e}} + c_1) \mathbf{1}_{e(x) \neq y} \mathbf{1}_{r(x) = \text{REMOTE}}. \quad (3)$$

As discussed previously, we assume that $m(x)$ is fixed while jointly training $r(x)$ and $e(x)$. The risk of equation 3 is defined as $R_{\text{general}} \triangleq \mathbf{E}_{(X,Y) \sim \mathcal{D}}[L_{\text{general}}(r, e, x, y; m)]$ and the *Bayes Classifiers for multi-class L2H* is defined as:

$$r^B, e^B \in \arg\min_{r,e} R_{\text{general}}(r, e; m). \quad (4)$$

The existence of these two Bayes classifiers will be proved in the next subsection.

## 2.2 BAYES OPTIMAL REJECTOR AND SERVER CLASSIFIER

In this subsection, we derive the Bayes classifiers as defined in equation 4 for both rejector and server classifier with a fixed client classifier $m(x)$ under generalized $0$-$1$ loss function in equation 3. As mentioned at the beginning of Sec. 2, the task we considered is multi-class classification. The client classifier $m(x)$ and server classifier $e(x)$ are multi-class classifiers with output to be one label among $K$ classes. The rejector $r(x)$ is a binary classifier with two possible labels: LOCAL and REMOTE, which means either making a prediction on the client or on the server. The rejector $r(x)$ will give a label LOCAL to $x$ if $r(x) > 0$ and give a label REMOTE to $x$ when $r(x) \leq 0$. Without loss of generality, we assume that for the Bayes classifier of rejector, $r^B(x) \in \{+1, -1\}$.

As discussed after equation 3, we consider the case where the client classifier $m$ is given and fixed during training. The output of the client classifier depends on input sample $x$, which can either be deterministic or stochastic, depending on the machine learning model of $m$. We formally consider the case where the client classifiers consist of $K$ sub-functions, say $[m_1(x), m_2(x), \cdots, m_K(x)]$. The $i$-th sub-function $m_i(x)$ represents the score for $i$-th class on sample $x$. Then the prediction of client classifier is $\arg\max_i m_i(x)$. For simplicity, we assume that $\arg\max_i m_i(x)$ is unique. Choosing the class that has the highest output score as prediction is a standard operation for multi-class classification. In the following analysis, we assume that the output of client is a random variable $M$ conditioned on the input $x$ (Neal, 2012).

**Theorem 2.1.** *Given a client classifier* $m(x)$, *the solutions of Bayes classifiers (defined in equation 4) for generalized* $0$-$1$ *loss under the space of all measurable functions are:*

$$e^B = \arg\max_i \eta_i(x), \quad (5)$$

*where* $\eta_i(x)$ *is defined in equation 2 and*

$$r^B = \mathbf{1}[\eta_{j^*(x)}(x) > (1 - c_e - c_1) + c_1 \max_i \eta_i(x)] \cdot 2 - 1, \quad (6)$$

*where* $j^*(x) \triangleq \arg\max_j m_j(x)$.

The proof is given in Appendix C. By Theorem 2.1, we find that the Bayes classifier for the servers $e^B$ is the same as the Bayes classifier for single classifiers (right side of equation 5). Also, the Bayes classifier for rejector $r^B$ compares the posterior risk (expected loss) for two different decisions: predicting on the client classifier or predicting on the server classifier. The function $\eta_{j^*(x)}(x)$ is the regression function for class predicted by client classifier $m(x)$, while $\max_i \eta_i(x)$ is the regression

function of the class predicted by server classifier. A larger value of the regression function tends to pull a sample to the classifier corresponding to the $\eta_i(x)$ because the regression function here means how likely that a sample belongs to the class predicted either by $m(x)$ or $e^B(x)$. If we write the right-hand side of the inequality inside the indicator (from equation 6) in another form: $(1 - c_e - c_1) + c_1 \max_i \eta_i(x) = 1 - c_e - c_1(1 - \max_i \eta_i(x))$, it's clear that larger $c_e$ and $c_1$ would impede reject decision. This result coincides with the intuition in scenarios we are interested in: if inquiring servers are likely to be more costly, we tend to finish tasks without asking for help too often.

However, we cannot directly get the Bayes classifiers $(e^B, r^B)$ in real-world tasks. To see this, in reality, we don't have the knowledge of the distribution $\mathcal{D}$ of the data set. Furthermore, we cannot approach the Bayes optimal classifiers through gradient-based methods because the generalized 0 - 1 loss (equation 3) is not differentiable since it contains variables in the indicator function. To solve this problem, we propose a surrogate loss function in Sec. 3. We show that the surrogate loss function is differentiable and convex, and optimal solutions of this surrogate loss function are consistent with the Bayes optimal classifiers.

## 3 STAGE-SWITCHING SURROGATE LOSS FUNCTION

In Sec. 2.2, the Bayes classifiers we derived minimize the risk of generalized 0 - 1 loss (equation 3). Since the distribution of the data set is unknown and the loss function is not differentiable, it's computationally intractable to get $(e^B, r^B)$ by solving equation 4. Another potential concern comes from the framework of the client-server system. In this system, the rejector is placed on the client side while the server classifier is placed on the server side, which requires synchronous updates between two sides while searching for the solution of the problem equation 4. In scenarios where the connection between the client and server is unstable, or bandwidth is limited as discussed in Sec. 1, continuously synchronizing the client and server during training becomes costly. These conditions necessitate that both the rejector and server classifier be capable of being trained in an asynchronous setting.

Similar with the definition of client classifier $m(x)$ in Sec. 2.2, we consider the case where server classifier $e(x)$ also consists of $K$ sub-functions, denoted as $[e_1(x), e_2(x), \cdots, e_K(x)]$. The prediction of the server classifier is $\arg\max_i e_i(x)$ and assumed to be unique. We further consider the case that the rejector consists of 2 sub-functions, say $[r_1(x), r_2(x)]$, where $r_1$ stands for the score of LOCAL while $r_2$ stands for the score of REMOTE. In accordance with the space of $r^B$ as stated in Sec. 2.2, the output of the rejector is defined as $r(x) \triangleq \mathbf{1}[r_1(x) > r_2(x)] \cdot 2 - 1$. Based on the definitions stated above, we propose a *stage-switching* surrogate loss function, which is differentiable and can be used in both synchronous and asynchronous settings. The surrogate loss function is defined as:

$$L_{\mathrm{S}}(r, e, x, y; m) = L_1(e, x, y) + L_2(r, e, x, y; m) \tag{7}$$

where

$$L_1(e, x, y) = -\ln \frac{\exp(e_y(x))}{\sum_{j=1}^{K} \exp(e_j(x))}, \tag{8}$$

and

$$L_2(r, e, x, y; m) = -(1 - c_e - c_1 + c_1 \mathbf{1}_{e(x)=y}) \ln \frac{\exp(r_2(x))}{\exp(r_2(x)) + \exp(r_1(x))} \tag{9}$$

$$- \mathbf{1}_{m(x)=y} \ln \frac{\exp(r_1(x))}{\exp(r_2(x)) + \exp(r_1(x))}.$$

Similarly, the surrogate risk is defined as

$$R_{\mathrm{S}} \triangleq \mathbf{E}_{(X,Y) \sim \mathcal{D}}[L_{\mathrm{S}}(r, e, x, y; m)]. \tag{10}$$

The surrogate loss function is designed as the summation of two sub-loss functions, $L_1$ and $L_2$ [1]. In the training process, we iteratively update the rejector or the server classifier while keeping the other

---

[1] Minimizing $R_{\mathrm{S}}$ now becomes an additive composite optimization problem (Boyd et al., 2011; He & Yuan, 2012; Nesterov, 2013; Li & Han, 2022).

fixed under each sub-loss function at one stage. Specifically, in *server stage*, we update the parameters of the server classifier under the sub-loss function $L_1$ while keeping the rejector unchanged, and in *rejector stage*, we update the parameters of rejector $r$ under sub-loss function $L_2$ while keeping server classifier unchanged. The way of switching stages depends on specific synchronous or asynchronous settings. Notice that the client classifier $m(x)$ is always fixed in all stages in the L2H framework. Detailed comparison with previous surrogate loss function for binary L2H is in Appendix A and comparison with direct extensions from other proposed surrogate loss functions in LWA and L2D is attached in Appendix B. The results show that previous works can not fully cover the settings that we are interested in.

Based on the stage-switching training method, the surrogate loss function is differentiable in all stages. In the server stage, the sub-surrogate loss function $L_1$ is the same as the cross-entropy loss function. In the rejector stage, for each given input sample $x$ with true label $y$, the indicator $\mathbf{1}_{e(x)=y}$ and $\mathbf{1}_{e(x)=y}$ are constants, that is, either one or zero. In either case, $L_2$ is differentiable since only the $r_1$ and $r_2$ are variables in this stage, and we can calculate the gradient with respect to $r_1$ and $r_2$. What's more, both $L_1$ and $L_2$ are convex w.r.t. $e$ as shown in the following subsection.

### 3.1 CONVEXITY AND MONOTONICITY OF THE SURROGATE LOSS FUNCTION

In the following proposition, we show that in both stages, the sub-surrogate loss function $L_1$ and $L_2$ are both convex or monotone.

**Proposition 3.1.** *For each given $(x, y)$, the loss function $L_1$ is convex over $e_i(x)$, for any $i \in [K]$; and the loss function $L_2$ is:*

- *convex over $r_1(x)$ and $r_2(x)$, when $1 - c_e - c_1 + c_1 \mathbf{1}_{e=y} > 0$;*

- *monotonically decreasing over $r_1$ and monotonically increasing over $r_2$ when $1 - c_e - c_1 + c_1 \mathbf{1}_{e=y} \leq 0$.*

The proof is given in Appendix D. The convexity of a function ensures that we can find the global minimizer through gradient-related optimization methods (Boyd & Vandenberghe, 2004; Nesterov, 2014). As for the monotonicity of $L_2$ when $1 - c_e - c_1 + c_1 \mathbf{1}_{e=y} \geq 0$, we will show that in Theorem 3.2, this property can help for solving the corner case in the proof. Based on differentiability, convexity, or monotonicity, we prove in Sec. 3.2 that our proposed surrogate loss function is consistent, that is, any minimizer of the surrogate risk (equation 10) also minimizes the generalized 0 - 1 risk of equation 3, referring to the consistency defined in Section 2.2 by Cao et al. (2023b).

### 3.2 CONSISTENCY OF SURROGATE LOSS FUNCTION

In this subsection, we verify the consistency between the surrogate loss function (equation 7) and the generalized 0 - 1 loss function (equation 3). Formally, we prove the following theorem:

**Theorem 3.2.** *Under the space of all measurable functions, the surrogate loss function (equation 7) is consistent with the generalized 0 - 1 loss function (equation 3), that is, the minimizer of the risk of surrogate loss function also minimizes the risk of original loss function:*

$$r^*, e^* \in \arg\min_{r,e} R_{\text{general}}(r, e; m),$$

*for all $r^*, e^* \in \arg\min_{r,e} R_{\text{S}}(r, e; m)$.*

The proof is given in Appendix E. As shown in Theorem 3.2, the function spaces for server classifier $e$ and rejector $r$ are both spaces of all measurable functions; training our model with this stage-switching surrogate loss function would eventually lead to Bayes classifiers, which is the global optimal solution.

In summary, we propose a stage-switching surrogate loss function, which is differentiable, convex, or monotone, and theoretically prove that this surrogate loss function is consistent. Based on those properties of our surrogate loss function, practical optimization methods can be conducted to derive the empirical minimizer of surrogate risk $R_{\text{S}}$, as defined in equation 10, with a finite number of data samples in real-world scenarios. In the next section, we propose computationally feasible algorithms for different settings PPR, IA, and BRR discussed in Sec. 1.

# 4 COMPUTATIONALLY FEASIBLE ALGORITHMS FOR LEARNING TO HELP

The stage-switching surrogate loss function that we propose in equation 7 ensures the usability of gradient-based optimization methods to train the rejector and server classifiers with flexibility in three settings we discuss in Section 1: PPR, IA, and BRR settings.

The training in the server stage is the same for all three settings since $L_1$ is only a function of the server classifier; no knowledge of the rejector or client classifier is needed. In the rejector stage, we set up different algorithms based on the constraints in different settings. Besides, for BRR settings, we propose post-hoc Algo. 4 and 5 after standard training. Since the stage-switching surrogate loss function is differentiable, any gradient-based optimization method can be used for training our model. To reduce the computation, we use Stochastic Gradient Descent (SGD) for presentation.

**Pay-Per-Request (PPR)**   Recall the scenarios we depict in Section 1, in the PPR setting, we consider that the client is always connected with the server; that is, the rejector has instant access to the latest version of server classifier $e(x)$ during the whole training phase. This setting works for scenarios where the connection is in real-time, and the bandwidth is ample and free during training, but each request conducts payment.

We design a synchronized algorithm, Algo. 1, that works for the PPR setting. The algorithm iteratively runs in the server stage, and in the rejector stage for each pair of samples and labels $(x, y)$. In the server stage, the server classifier $e(x)$ is updated by the sub-loss $L_1$. In the rejector stage, the current $e(x)$ is instantly shared with the rejector. Together with the outputs of the fixed client classifier and current server classifier, the sub-loss $L_2$ is calculated to update the rejector.

---

**Algorithm 1** Optimization With Our Surrogate Loss Function

---

**Input:** Training set $\{(x_i, y_i) : i \in [n]\}$, Fixed client classifier $m$, Rejector $r^0$, Sever classifier $e^0$, Constant cost $c_1$ and $c_e$.

1: **for** $t = 1$ to $n$ **do**

2:     $L_1^t(x_t, y_t) = -\ln \frac{\exp(e_{y_t}(x_t))}{\sum_j \exp(e_j(x_t))}$

3:     $e^t \leftarrow \texttt{SGD}(L_1^t(x_t, y_t), e^{t-1})$

4:     $L_2^t(x_t, y_t) = -\mathbf{1}_{m(x_t)=y_t} \ln \frac{\exp(r_1(x_t))}{\exp(r_2(x_t))+\exp(r_1(x_t))} - (1 - c_e - c_1 + c_1 \mathbf{1}_{e^t(x_t)=y_t}) \ln \frac{\exp(r_2(x_t))}{\exp(r_2(x_t))+\exp(r_1(x_t))}$

5:     $r^t \leftarrow \texttt{SGD}(L_2(x_t, y_t), r_{t-1})$

6: **end for**

7: **return** $r^n, e^n$

---

**Intermittent Availability (IA)**   In the IA setting, the connection between the client and the server is not always available. This scenario may happen in any distributed system that is connected to the public network because of traffic control. Also in private networks, connection may be lost due to instability of the network. If we still train our model with synchronized algorithm 1, the training process will be postponed once the connection is delayed or lost. Holding the status of both the rejector and server classifier will cause a waste of resources and time.

Our stage-switching surrogate loss function in equation 7 consists of two sub-loss functions. An asynchronized algorithm is best to tackle those issues. The idea is to separately train the server classifier with $L_1$ on the server side and train the rejector with $L_2$ while keeping a temporary version of the server classifier $e^-$ on the client side. The $e^-$ will be updated only when the intermittent connection between client and server is built. Considering that the client and server only connect every $S$ time slot we propose the asynchronized training algorithm in Algo. 2. In line 4 of Algo. 2, the server connects with the client and sends the current server classifier $e$ as the latest $e^-$. In line 5, we calculate the $L_2$ loss with stored server classifier $e^-$.

**Bounded-Reject-Rate (BRR)**   While a few works focus on the BRR setting in L2D and L2H frameworks, the setting is indeed realistic. For example, subscribers of `ChatGPT` or other AI assistants don't pay extra money for each inquiry to advanced models, but they are allowed to only

---

**Algorithm 2** Asynchronous Optimization With Our Surrogate Loss Function

---

**Input:** Training set $\{(x_i, y_i) \mid i \in [n]\}$, Fixed client classifier $m$, Rejector $r^0$, Sever classifier $e^0$, Constant cost $c_1$ and $c_e$, Synchronization interval $S$, .

1: **for** $t = 1$ to $n$ **do**
2:     $L_1^t(x_t, y_t) = -\ln \frac{\exp\left(e_{y_t}(x_t)\right)}{\sum_j \exp\left(e_j(x_t)\right)}$
3:     $e^t \leftarrow \text{SGD}(L_1^t(x_t, y_t), e^{t-1})$
4:     **if** $t \bmod S = 1$ **then** $e^- \leftarrow e^t$ **end if**
5:     $L_2^t(x_t, y_t) = -(1 - c_e - c_1 + c_1 \mathbf{1}_{e^-(x_t) = y_t}) \ln \frac{\exp\left(r_2(x_t)\right)}{\exp\left(r_2(x_t)\right) + \exp\left(r_1(x_t)\right)} -$
    $\mathbf{1}_{m(x_t) = y_t} \ln \frac{\exp\left(r_1(x_t)\right)}{\exp\left(r_2(x_t)\right) + \exp\left(r_1(x_t)\right)}$
6:     $r^t \leftarrow \text{SGD}(L_2(x_t, y_t), r_{t-1})$
7: **end for**
8: **return** $r^n, e^n$

---

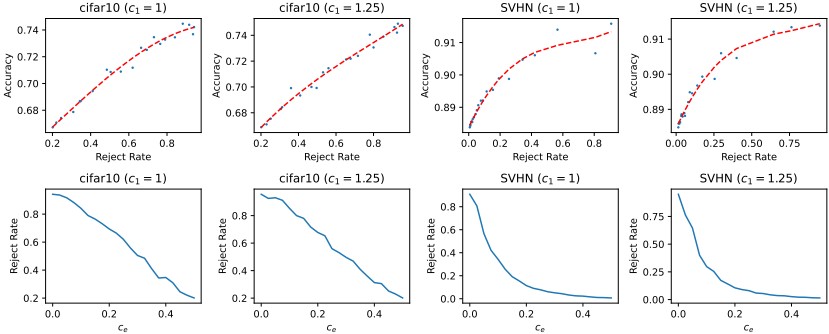

Figure 2: Impact of $c_e$ nad $c_1$ on accuracy and reject rate. First row: change of accuracy as reject rate changes; Second row: change of reject rate as reject cost $c_e$ changes.

ask an assigned number of questions each hour during peak hours. In this setting, we consider that making a reject decision and sending samples to the server is free, but the number of samples that can be sent to the server in units of time, i.e., reject rate, has a fixed upper bound $q$. There are two concerns in this setting: a) the rejector should actively pick up the samples that have higher accuracy on the server classifier compared to the client classifier; b) the actual reject rate in the inference phase should be bounded.

To address the concerns stated above, we propose stochastic post-hoc algorithms in Algo. 3, 4, and 5. These algorithms sample random variables to decide either sending extra samples to server when empirical reject rate is lower than upper bound $q$ or keep a portion of rejected samples locally when it's higher than upper bound $q$. By conducting this stochastic post-hoc algorithm, we can ensure the bounded reject rate while making the best use of server classifier. The detailed discussions and algorithms are in Appendix F.

## 5 EXPERIMENT

In this section, we test the proposed surrogate loss function in equation 7 and algorithms for different settings on CIFAR-10 (Krizhevsky & Hinton, 2009), SVHN (Netzer et al., 2011) , and CIFAR-100 (Krizhevsky & Hinton, 2009) datasets.

In our experiments, the base network structure for the client classifier and the rejector is LeNet-5, and the server classifier is either AlexNet or ViT. The client classifier is trained solely and works as a fixed model in the following process. Since Algo. 1 is the basic algorithm for our multi-class L2H framework, we start from the evaluation for the PPR setting.

**Trade-off accuracy and reject rate on reject cost and inaccuracy cost** In this experiment, we evaluate the impact of reject cost $c_e$ and inaccuracy cost $c_1$ on the overall accuracy and reject rate,

Table 1: Contrastive Evaluation Results with $c_1 = 1.25$ and $c_e = 0.25$

|  | cifar10 (%) | | | | SVHN (%) | | | |
|---|---|---|---|---|---|---|---|---|
|  | ratio | m | e | differ. | ratio | m | e | differ. |
| data with $r(x) = $ LOCAL | 44.11 | 73.9 | 81.9 | **8.0** | 91.71 | 90.6 | 93.3 | **2.7** |
| data with $r(x) = $ REMOTE | 55.9 | 54.5 | 67.7 | **13.2** | 8.29 | 61.2 | 72.8 | **11.6** |

Figure 3: Comparison of the training loss for different synchronized settings.

respectively. Specifically, we choose $c_e$ from an interval between $[0, 0.5]$ with fixed inaccuracy costs $c_1 = 1$ and $c_1 = 1.25$. From Fig. 2 we see that, incorporating a server classifier indeed helps increase the overall accuracy, while both the cost of rejection and inaccuracy on the server will balance the usage of the client classifier and server classifier.

**Contrastive evaluation over client and server classifiers**     To see what the rejector learns in training and how it works in the inference phase, we conduct a contrastive evaluation, where in the inference phase, we split the testing set into a rejected subset and non-rejected subset, according to the output of $r(x)$. We test the accuracy of the rejected and non-rejected subset on both the client classifier $m(x)$ and the server classifier $e(x)$. The result is shown in Table 1, which implies that the rejector mostly only sends the samples that are predicted inaccurately on $m(x)$ while predicted more accurately on $e(x)$ to the server end. Column "ratio" indicates the percentage of samples over dataset have the same $r(x)$. For different $c_e$ and $c_1$, the results are similar as shown in Appendix G.

**Convergence rate comparison over synchronization and asynchronization**     We compare the convergence rate and accuracy with models trained by Algo. 1 for PPR and that trained by Algo. 2 for IA. The accuracy performances are similar in both algorithms, as compared Table 1 with Table 7 in Appendix G. This result coincides with the comparison in Fig. 3, where a larger synchronization interval $S$ causes a slower convergence rate but converges to the same level of loss. The experiment results are in line with our expectation since the minimizers of our surrogate loss function derived in Theorem 3.2, are independent to specific optimization algorithms.

Experiments for stochastic post-hoc Algorithm 3, 4, and 5, its comparison with randomly reject under BRR settings, experiments on CIFAR-100 with server classifier to be ViT, together with other additional experiment materials, are in Appendix G.

## 6 CONCLUSION

In this paper, we extend the Learning to Help framework to address multi-class classification with different resource constraints: PAY-PER-REQUEST, INTERMITTENT AVAILABILITY, and BOUNDED REJECT RATE. By introducing a stage-switching surrogate loss function, we enabled effective training of both the server-side classifier and the rejector with computationally feasible algorithms for three settings. Our experimental results demonstrate that the proposed methods offer a practical and efficient solution for multi-class classification, aligning with the theoretical guarantees we established. This work opens new opportunities for further exploration of multi-party collaboration in hybrid machine learning systems, especially in the era of LLM.

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

## A  COMPARISON WITH BINARY SURROGATE LOSS

In this section, we compare the surrogate loss function proposed in binary case by Wu & Sarwate (2024) with our proposed method in Section 3.

In this work, the formula of the surrogate loss function is:

$$L_{\text{Wu}}(r, e, x, y) = c_1 \exp\left(\frac{\beta}{2}(-ey - r)\right) + c_e \exp(-r) + 1_{m(x)y \leq 0} \exp(\alpha r)$$

where $\alpha$ and $\beta$ are parameters for calibration. If we directly make the extension to multi-class in our setting, we can keep the same definition of rejector while the output of client classifier and server classifier becomes $\arg\max_i m_i(x)$ and $\arg\max_i e_i(x)$, which results in the surrogate loss:

$$L'_{\text{Wu}}(r, e, x, y) = c_1 \exp\left(\frac{\beta}{2}(-\arg\max_i e_i(x)y - r)\right) + c_e \exp(-r) + 1_{\arg\max_i m_i(x)y \leq 0} \exp(\alpha r)$$

The client classifier part works well since the client classifier is fixed during training. However, this part $\arg\max_i m_i(x)y$ will lead to trouble since both $\arg\max_i m_i(x)$ and $y$ are index numbers of class in multi-class and therefore, the function is not differentiable with respect to $e_i(x)$.

Furthermore, this surrogate loss function requires to calculate calibration parameters $\alpha$ and $\beta$, which depends on the estimation of the accuracy of the client classifier, for each input sample $x$ as shown in Theorem 2 in this work (Wu & Sarwate, 2024), which adds more calculations and estimation errors to practical experiments. This surrogate loss function is not available in the IA setting since it's jointly training the rejector and server classifier at the same time.

Compared to the extension of the surrogate loss function proposed by Wu & Sarwate (2024) with our stage-switching surrogate loss function, our method is differentiable in multi-class and more flexible for fitting different settings.

## B  COMPARISON WITH DIFFERENT SURROGATE LOSS FUNCTIONS

In this section, we justify that previous surrogate loss functions, proposed from LWA and L2D, can not be directly extended to L2H by comparing those surrogate loss functions with our method, shown in Table 2, on the properties that are needed in the settings that we are interested in.

Confidence-LWA (or score-based-LWA) represents the methods in LWA that use the output $m(x)$ or function of output $f(m(x))$ of the client classifier as a metric of confidence on current prediction. If the value of the metric is smaller than a threshold, then the input sample $x$ will be discarded as a result of rejection. The theoretical framework of Confidence-LWA was first proposed by Herbei & Wegkamp (2006). There is no separate rejector in those methods. The rejected samples will be abandoned since there will be no more actions after rejection. Separate rejector-LWA represents those surrogate loss functions in LWA that assume a separate rejector to independently decide whether to reject or not. This kind of method was first proposed by Cortes et al. (2016) and proves the inability of confidence-based methods in special cases, referring to Sec. 2.2 of this work. However, in the LWA framework, the rejection decision only depends on the local classifier $m$ and does not consider the issues after rejection.

In the L2D framework, samples are sent to an expert after rejection. Softmax-L2D, firstly proposed by Mozannar & Sontag (2020b), and OvA-L2D, firstly proposed by Verma & Nalisnick (2022) are two different classes of surrogate loss functions. Softmax-L2D adds a reject option as another class of task to the cross-entropy loss function, while OvA-L2D uses surrogate loss functions that transform multi-class classification into a combination of multi-binary classifiers. However, the output of the rejector on both methods also depends on the output of each class $m_i(x)$ on mobile classifier $m(x)$. Moreover, during training, the client and the server must keep connection with each other.

Two-stage (TS) confidence-L2D and two-stage (TS) separate rejector-L2D are those kinds of surrogate loss functions that consist of different sub-loss functions and can be trained in corresponding stages for both confidence and separate rejector cases, firstly proposed by Mao et al. (2023). Those two methods can be used for IA settings where the training is asynchronized since their surrogate loss

function can be easily split and run in different locations. However, in two-stage confidence-L2D methods, the rejector takes the output of $m(x)$ as input, while in two-stage separate rejector-L2D methods, the rejector only considers the performance of experts on different servers.

For those methods that do not have a separate rejector, the limitation has been proved by Cortes et al. (2016). What's more, usually ML classifiers are more complex than rejectors in scale. In the scenarios of our PPR and IA settings, when the latency is a concern, a system with a separate reject is faster since in a non-separate rejector system, all samples are firstly sent to the client classifier and then decided by the rejector, which may take longer time.

For those methods that the rejector does not learn from client classifier $m(x)$ during training, the reject decision only depends on the performance of server classifier $e(x)$. For those methods that the rejector does not learn from server classifier $e(x)$, the reject decision only depends on the performance of client classifier $m(x)$. In our PPR and IA settings, asking the server for help is not free, so we have to balance the use of the server. A desired rejector should learn the ability to compare the performance of $m(x)$ and $e(x)$ for each input $x$ as shown in the Bayes classifier by 2.1.

Those methods that can not train in asynchronized fields may fail in the IA setting since the connection between client and server is unstable and intermittent.

Based on the settings that are realistic and interesting in real-world scenarios, we require a surrogate loss function for a multi-class L2H system to have a separate rejector, and the rejector learns from the knowledge of both the client classifier and server classifier during training and can operate in asynchronized fashion. Previous works that stem from LWA and L2H can not be directly extended to L2H because they are either technically non-differentiable for training server classifiers or conceptually unmatched with the required properties stated above.

Table 2: Difference among surrogate loss function proposed in LWA and L2D

| | Separate rejector | Rejector learns $m(x)$ | Rejector learns $e(x)$ | Async |
|---|---|---|---|---|
| Confidence-LWA | × | ✓ | × | - |
| Separate rejector-LWA | ✓ | ✓ | × | - |
| Softmax-L2D | × | ✓ | ✓ | × |
| OvA-L2D | × | ✓ | ✓ | × |
| TS confidence-L2D | × | ✓ | ✓ | ✓ |
| TS separate rejector-L2D | ✓ | × | ✓ | ✓ |
| Ours | ✓ | ✓ | ✓ | ✓ |

## C    PROOF OF THEOREM 2.1

The standard approach to deriving the Bayes classifier for any given loss function is to compare the posterior risk associated with each possible decision of the classifier. The decision that minimizes the posterior risk is then chosen as the output. Recall in Section 2.1, the regression function is $\eta_i(x) = P(Y = i|X = x)$ and the generalized $0$-$1$ loss function defined in equation 3 is:

$$L_{\text{general}}(r, e, x, y; m) = \mathbf{1}_{m(x) \neq y} \mathbf{1}_{r(x)=\text{LOCAL}}$$
$$+ c_{\text{e}} \mathbf{1}_{e(x)=y} \mathbf{1}_{r(x)=\text{REMOTE}} + (c_{\text{e}} + c_1) \mathbf{1}_{e(x) \neq y} \mathbf{1}_{r(x)=\text{REMOTE}}.$$

Note that $m(x)$ is also stochastic, as discussed in Section 2.2. The random variable corresponding to the client classifier is denoted by $M$. Then, for any given sample $x$, the posterior risk over the conditional distribution of $Y$ and $M$ of any decision $r' \in \{\text{REMOTE}, \text{LOCAL}\}$ and $e' \in [K]$ is:

$$\mathbb{E}_{y,m|x} L_{\text{general}}(r', e', x, y; m)$$
$$= \mathbb{E}_{y,m|x}(\mathbf{1}_{m(x) \neq y} \mathbf{1}_{r'=\text{LOCAL}} + c_{\text{e}} \mathbf{1}_{e'=y} \mathbf{1}_{r'=\text{REMOTE}} + (c_{\text{e}} + c_1) \mathbf{1}_{e' \neq y} \mathbf{1}_{r'=\text{REMOTE}})$$
$$= \mathbb{E}_{m|x}(P(Y = e'|X = x)(\mathbf{1}_{m(x) \neq e'} \mathbf{1}_{r'=\text{LOCAL}} + c_{\text{e}} \mathbf{1}_{e'=e'} \mathbf{1}_{r'=\text{REMOTE}})$$
$$+ P(Y \in \mathcal{Y}_{e'}|X = x)(\mathbf{1}_{m(x) \notin \mathcal{Y}_{e'}} \mathbf{1}_{r'=\text{LOCAL}} + (c_{\text{e}} + c_1) \mathbf{1}_{e' \notin \mathcal{Y}_{e'}} \mathbf{1}_{r'=\text{REMOTE}})),$$

where $\mathcal{Y}_{e'} = \mathcal{Y} \setminus \{e'\}$ is the set of all labels except for $e'$. Then,

$$
\begin{aligned}
\mathbb{E}_{y,m|x} & L_{\text{general}}(r', e', x, y; m) \\
&= \mathbb{E}_{m|x}(\eta_{e'}(x)(\mathbf{1}_{m(x) \neq e'} \mathbf{1}_{r'=\text{LOCAL}} + c_e \mathbf{1}_{r'=\text{REMOTE}}) \\
&\quad + (1 - \eta_{e'}(x))(\mathbf{1}_{m(x) \notin \mathcal{Y}_{e'}} \mathbf{1}_{r'=\text{LOCAL}} + (c_e + c_1)\mathbf{1}_{r'=\text{REMOTE}})) \\
&= \mathbb{E}_{m|x}(\mathbf{1}_{r'=\text{LOCAL}} \left(\eta_{e'}(x)\mathbf{1}_{m(x) \neq e'} + (1 - \eta_{e'}(x))\mathbf{1}_{m(x) \notin \mathcal{Y}_{e'}}\right) \\
&\quad + \mathbf{1}_{r'=\text{REMOTE}} \left(c_e \eta_{e'}(x) + (c_e + c_1)(1 - \eta_{e'}(x))\right)) \\
&= \mathbb{E}_{m|x}(\mathbf{1}_{r'=\text{LOCAL}}(\eta_{e'}(x)\mathbf{1}_{m(x) \neq e'} + (1 - \eta_{e'}(x))\mathbf{1}_{m(x) \notin \mathcal{Y}_{e'}}) \\
&\quad + \mathbf{1}_{r'=\text{REMOTE}}(c_e + c_1(1 - \eta_{e'}(x)))) \\
&= \mathbf{1}_{r'=\text{LOCAL}}(\eta_{e'}(x)P(M \neq e' \mid X = x) + (1 - \eta_{e'}(x))P(M \notin \mathcal{Y}_{e'} \mid X = x)) \\
&\quad + \mathbf{1}_{r'=\text{REMOTE}}(c_e + c_1(1 - \eta_{e'}(x))) \\
&= \mathbf{1}_{r'=\text{LOCAL}}P(M \neq Y \mid X = x) + \mathbf{1}_{r'=\text{REMOTE}}(c_e + c_1(1 - \eta_{e'}(x))).
\end{aligned}
\tag{11}
$$

Also, for any $e'$, the following inequality holds:

$$
\begin{aligned}
\mathbf{1}_{r'=\text{LOCAL}}P(M \neq Y \mid X = x) &+ \mathbf{1}_{r'=\text{REMOTE}}(c_e + c_1(1 - \eta_{e'}(x))) \\
&\geq \mathbf{1}_{r'=\text{LOCAL}}P(M \neq Y \mid X = x) + \mathbf{1}_{r'=\text{REMOTE}}(c_e + c_1(1 - \max_i \eta_i(x)))
\end{aligned}
\tag{12}
$$

The equality in Inequality 12 holds if and only if the decision $e'$ satisfies $\eta_{e'}(x) = \max_i \eta_i(x)$. By combining this with equation 11, the Bayes classifier $e^B$ defined in equation 4, which minimizes $\mathbb{E}_{y,m|x} L_{\text{general}}(r', e', x, y; m)$, satisfies $\eta_{e^B} = \max_i \eta_i(x)$. Therefore, the Bayes classifier for the local client is:

$$
e^B = \arg\max_i \eta_i(x).
$$

Recalling the lower bound given by equation 12, the following lower bound can be derived:

$$
\begin{aligned}
\mathbf{1}_{r'=\text{LOCAL}}P(M \neq Y \mid X = x) &+ \mathbf{1}_{r'=\text{REMOTE}}(c_e + c_1(1 - \max_i \eta_i(x)) \\
&\geq \min\{P(M \neq Y \mid X = x), c_e + c_1(1 - \max_i \eta_i(x)\}
\end{aligned}
\tag{13}
$$

Inequality 13 implies that for a given $x$, if $P(M \neq Y \mid X = x) \geq c_e + c_1(1 - \max_i \eta_i(x))$, the posterior risk reaches the lower bound when $r' = \text{REMOTE}$. Conversely, if $P(M \neq Y \mid X = x) < c_e + c_1(1 - \max_i \eta_i(x))$, the posterior risk reaches the lower bound when $r' = \text{LOCAL}$. By the definition of the rejector, one way to construct the Bayes classifier of the rejector is:

$$
\begin{aligned}
r^B &= \mathbf{1}[P(M \neq Y | X = x) < c_e + c_1(1 - \max_i \eta_i(x))] \cdot 2 - 1 \\
&= \mathbf{1}[1 - P(M = Y | X = x) < c_e + c_1(1 - \max_i \eta_i(x))] \cdot 2 - 1 \\
&= \mathbf{1}[1 - \eta_{\arg\max_j m_j(x)}(x) < c_e + c_1(1 - \max_i \eta_i(x))] \cdot 2 - 1 \\
&= \mathbf{1}[\eta_{\arg\max_j m_j(x)}(x) > (1 - c_e - c_1) + c_1 \max_i \eta_i(x)] \cdot 2 - 1.
\end{aligned}
$$

In summary, the Bayes classifiers for 0-1 generalized loss are

$$
e^B = \arg\max_i \eta_i(x)
$$

and

$$
r^B = \mathbf{1}[\eta_{j^*(x)}(x) > (1 - c_e - c_1) + c_1 \max_i \eta_i(x)] \cdot 2 - 1,
$$

where $j^*(x) \triangleq \arg\max_j m_j(x)$.

## D    PROOF OF PROPOSITION 3.1

We first show the convexity of $L_1$ and $L_2$ w.r.t $e$. For $L_1$ in equation 8, we notice that it equals to cross-entropy loss function for multi-class tasks. Referring to the proof by Cover & Thomas (2005), in which the cross-entropy is transformed into the form of KL divergence, which is convex, we can

directly draw the conclusion that the $L_1$ is a convex function over $e_i(x)$ for any $i \in [K]$ given $(x, y)$.

Next, we prove the convexity of $L_2$ w.r.t $e$. Recall the definition of $L_2$ in equation 9:

$$L_2 = -(1 - c_e - c_1 + c_1 \mathbf{1}_{e=y}) \ln \frac{\exp(r_2)}{\exp(r_2) + \exp(r_1)} - \mathbf{1}_{m=y} \ln \frac{\exp(r_1)}{\exp(r_2) + \exp(r_1)}.$$

As described in Section 3, once in the rejector stage, only the rejector $r$ is variable. For convenience, we rewrite $L_2$ in this form:

$$f(r_1, r_2) = -a \ln \frac{\exp(r_2)}{\exp(r_2) + \exp(r_1)} - b \ln \frac{\exp(r_1)}{\exp(r_2) + \exp(r_1)},$$

where $a = 1 - c_e - c_1 + c_1 \mathbf{1}_{e=y}$ and $b = \mathbf{1}_{m=y}$, which are depends on $(x, y)$. The analysis can be divided into two cases:

- case (1): $a \leq 0$,
- case (2): $a > 0$.

In case (1), the partial derivative of $f(r_1, r_2)$ are:

$$\frac{\partial f(r_1, r_2)}{\partial r_1} = a \frac{\exp(r_1)}{(\exp(r_2) + \exp(r_1))} - b \frac{\exp(r_2)}{(\exp(r_2) + \exp(r_1))},$$

and

$$\frac{\partial f(r_1, r_2)}{\partial r_2} = -a \frac{\exp(r_1)}{(\exp(r_2) + \exp(r_1))} + b \frac{\exp(r_2)}{(\exp(r_2) + \exp(r_1))}.$$

Since $b = \{+1, -1\}$, we have that:

$$\frac{\partial f(r_1, r_2)}{\partial r_1} \leq a \frac{\exp(r_1)}{(\exp(r_2) + \exp(r_1))} \leq 0,$$

and

$$\frac{\partial f(r_1, r_2)}{\partial r_2} \geq -a \frac{\exp(r_1)}{(\exp(r_2) + \exp(r_1))} \geq 0$$

always holds, which means $f(r_1, r_2)$ (or $L_2$), are non-increasing function of $r_1$ and non-decreasing function of $r_2$. For the case $c_e > c_1 = 1$, it is impossible for $a = 0$. Thus, $f(r_1, r_2)$ (or $L_2$), are monotonically decreasing function of $r_1$ and monotonically increasing function of $r_2$.

In case (2), where $a > 0$, we have that $a + b > 0$ always holds. Then we calculate $(f(r_1, z_1) + f(r_1, z_2))/2$ and $f(r_1, (z_1 + z_2)/2)$ for any $z_1$ and $z_2$:

$$
\begin{aligned}
(f(r_1, &z_1) + f(r_1, z_2))/2 \\
&= \frac{1}{2}(-a(z_1 - \ln(e^{z_1} + e^{r_1})) \\
&\quad - b(r_1 - \ln(e^{r_1} + e^{z_1})) - a(z_2 - \ln(e^{z_2} + e^{r_1})) - b(r_1 - \ln(e^{r_1} + e^{z_2}))) \\
&= \frac{1}{2}(-az_1 + a\ln(e^{z_1} + e^{r_1}) - 2br_1 \\
&\quad + b\ln(e^{r_1} + e^{z_1}) - az_2 + a\ln(e^{z_2} + e^{r_1}) + b\ln(e^{z_2} + e^{r_1})) \\
&= \frac{1}{2}(-a(z_1 + z_2) - 2br_1 + (a + b)\ln(e^{z_1 + z_2} + e^{r_1 + z_2} + e^{r_1 + z_1} + e^{2r_1})),
\end{aligned}
$$

and

$$
\begin{aligned}
f(r_1, (z_1 + z_2)/2) &= -a((z_1 + z_2)/2 - \ln(e^{(z_1 + z_2)/2} + e^{r_1})) - b(r_1 - \ln(e^{r_1} + e^{(z_1 + z_2)/2})) \\
&= \frac{-a(z_1 + z_2)}{2} + a\ln(e^{(z_1 + z_2)/2} + e^{r_1}) - br_1 + b\ln(e^{r_1} + e^{(z_1 + z_2)/2}).
\end{aligned}
$$

The subtraction between them is:

$$(f(r_1, z_1) + f(r_1, z_2))/2 - f(r_1, (z_1 + z_2)/2) \tag{14}$$

$$= \frac{1}{2}(a + b) \ln(e^{z_1 + z_2} + e^{r_1 + z_2} + e^{r_1 + z_1} + e^{2r_1})$$

$$- (a + b) \ln(e^{(z_1 + z_2)/2} + e^{r_1})$$

$$= (a + b) \ln \frac{\sqrt{(e^{z_1} + e^{r_1})(e^{z_2} + e^{r_1})}}{e^{(z_1 + z_2)/2} + e^{r_1}}$$

We notice that exponential function $\exp(z)$ is convex function such that

$$\frac{e^{z_1} + e^{z_2}}{2} \geq e^{(z_1 + z_2)/2}$$

holds for any $z_1, z_2$. Since

$$\frac{e^x + e^y}{2} \geq e^{(x+y)/2}$$

$$\Leftrightarrow e^{r_1 + y} + e^{r_1 + x} \geq 2e^{(x+y)/2 + r_1}$$

$$\Leftrightarrow (e^x + e^{r_1})(e^y + e^{r_1}) \geq (e^{(x+y)/2} + e^{r_1})^2$$

$$\Leftrightarrow \frac{\sqrt{(e^x + e^{r_1})(e^y + e^{r_1})}}{e^{(x+y)/2} + e^{r_1}} \geq 1,$$

and $a + b > 0$, we prove that the for equation 14:

$$(f(r_1, z_1) + f(r_1, z_2))/2 - f(r_1, (z_1 + z_2)/2) \geq 0$$

always holds, that is $f(r_1, r_2)$ is a convex function over $r_2$. Following similar steps, we also prove that $f(r_1, r_2)$ is a convex function over $r_1$.

## E    PROOF OF THEOREM 3.2

The surrogate loss function defined in equation 7 consists of $L_1$ and $L_2$, which are targeted to the server classifier and rejector, respectively. Since the rejector and server classifier are trained in a stage-switching manner as described in Section 3, parameters of $e$ are only updated by the gradient derived from $L_1$, and that of $r$ is only updated by the gradient derived from $L_2$. Therefore, the minimizer of $e$ is derived from the risk of $L_1$ in the server stage, and the minimizer of $r$ is derived from the risk of $L_2$ (with the current server classifier) in the rejector stage. The risks are:

$$\mathbb{E}_{y,m|x} L_1 = -\sum_{y \in \mathcal{Y}} \eta_y(x) \log \left( \frac{\exp(e_y(x))}{\sum_{y' \in \mathcal{Y}} \exp(e_{y'}(x))} \right)$$

and

$$\mathbb{E}_{y,m|x} L_2 = -\mathbb{E}_{y,m|x}(1 - c_e - c_1 + c_1 \mathbf{1}_{e=y}) \ln \frac{\exp(r_2)}{\exp(r_2) + \exp(r_1)} - \mathbb{E}_{y,m|x} \mathbf{1}_{m=y} \ln \frac{\exp(r_1)}{\exp(r_2) + \exp(r_1)}$$

Since the Bayes classifiers are optimal for any $x$ according to the definition in equation 4, it's equivalent to deriving the minimizer of risk of each sub-surrogate loss function for any given $x$.

Note that $L_1$ is a cross-entropy loss function for server classifier $e(x)$; we can directly get the minimizers for each $e_i$ by the arguments in Section 10.6 proposed by Hastie et al. (2009). Let's denote the minimizers of each sub-function $e_i$, which is defined in Section 3, of server classifier by $e_i^*$. The minimizes $e_i^*$ satisfy the following condition:

$$\frac{\exp(e_i^*(x))}{\sum_j \exp(e_j^*(x))} = \eta_i(x), \quad \forall i,$$

where $\eta_i(x)$ is the regression function for $i$-th class. Therefore, the $e^*(x) = \arg\max_i e_i^*(x) = \arg\max_i \eta_i(x) = e^B(x)$. The minimizer of server classifier $e^*$ is the same as the Bayes classifier for server $e^B$.

Next, we consider the rejector $r(x)$. Since $e(x) = \arg\max_j e_j(x)$, when we plug in the minimizers $e_i^*(x)$ to $L_2$, the risk of $L_2$ conditioned on $x$ becomes:

$$\mathbb{E}_{y,m|x} L_2 = \mathbb{E}_{y,m|x}\left(-\mathbf{1}_{e=y}\ln\frac{\exp(r_2)}{\exp(r_2)+\exp(r_1)} - \mathbf{1}_{m=y}\ln\frac{\exp(r_1)}{\exp(r_2)+\exp(r_1)}\right)$$

$$= -(1 - c_e - c_1 + c_1\max_i\eta_i(x))\ln\frac{\exp(r_2)}{\exp(r_2)+\exp(r_1)}$$

$$- \eta_{j^*(x)}\ln\frac{\exp(r_1)}{\exp(r_2)+\exp(r_1)},$$

where $j^*(x) = \arg\max_i m_i(x)$ as defined in Theorem 2.1. The following analysis will be divided into two cases:

- case (a): $(1 - c_e - c_1) + c_1\max_i\eta_i(x) \le 0$,
- case (b): $(1 - c_e - c_1) + c_1\max_i\eta_i(x) > 0$.

In case (a), when $(1 - c_e - c_1) + c_1\max_i\eta_i(x) \le 0$, $\partial\mathbb{E}_{y,m|x}L_2/\partial r_1 \le 0$ always holds, such that $\mathbb{E}_{y,m|x}L_2$ is monotonically decreasing function of $r_1$. And $\partial\mathbb{E}_{y,m|x}L_2/\partial r_2 > 0$ always holds such that $\mathbb{E}_{y,m|x}L_2$ monotonically increasing function of $r_2$. In fact, we can also draw the same conclusions by using the monotonicity from Proposition 3.1 since the expectation of monotone function is still monotone. Therefore, in this case,

$$\frac{\exp(r_1^*)}{(\exp(r_2^*)+\exp(r_1^*))} \longrightarrow 1 \quad\text{and}\quad \frac{\exp(r_2^*)}{(\exp(r_2^*)+\exp(r_1^*))} \longrightarrow 0$$

That means $r_1^* > r_2^*$ always holds. Therefore, the rejector always satisfies:

$$r^*(x) = \mathbf{1}[r_1^*(x) > r_2^*(x)]\cdot 2 - 1 = 1$$

Also, in this case, the Bayes classifier for rejector is $r^B = \mathbf{1}[\eta_{\arg\max_i m_i(x)}(x) > (1 - c_e - c_1) + c_1\max_i\eta_i(x)]\cdot 2 - 1 = 1\cdot 2 - 1 = 1$. Therefore, $r^*(x) = r^B(x)$.

Then we consider the case (b) when $(1 - c_e - c_1) + c_1\max_i\eta_i(x) > 0$. Since $L_1$ is convex according to Proposition 3.1, and expectation of convex function is still convex proved by Boyd & Vandenberghe (2004), $\mathbb{E}_{y|x}L_2$ is convex, and its minimizers are achieved when partial derivative equal to $0$. We take the partial derivative over $r_1$ and $r_2$:

$$\frac{\partial\mathbb{E}_{y|x}L_2}{\partial r_1} = (1 - c_e - c_1 + c_1\max_i\eta_i(x))\frac{\exp(r_1)}{(\exp(r_2)+\exp(r_1))} - \eta_{j^*(x)}\frac{\exp(r_2)}{(\exp(r_2)+\exp(r_1))},$$

and

$$\frac{\partial\mathbb{E}_{y|x}L_2}{\partial r_2} = -(1 - c_e - c_1 + c_1\max_i\eta_i(x))\frac{\exp(r_1)}{(\exp(r_2)+\exp(r_1))} + \eta_{j^*(x)}\frac{\exp(r_2)}{(\exp(r_2)+\exp(r_1))}.$$

Setting the partial derivatives to $0$, we have that

$$\frac{\exp(r_1^*)}{\exp(r_2^*)} = \frac{\eta_{j^*(x)}(x)}{1 - c_e - c_1 + c_1\max_i\eta_i(x)}.$$

Then, the rejector becomes:

$$r^*(x) = \mathbf{1}[r_1^*(x) > r_2^*(x)]\cdot 2 - 1$$
$$= \mathbf{1}[\exp(r_1^*(x)) > \exp(r_2^*(x))])\cdot 2 - 1$$
$$= \mathbf{1}[\frac{\exp(r_1^*(x))}{\exp(r_2^*(x))} > 1])\cdot 2 - 1$$
$$= \mathbf{1}[\eta_{j^*(x)}(x) > (1 - c_e - c_1) + c_1\max_i\eta_i(x)]\cdot 2 - 1$$
$$= r^B(x).$$

We prove that in both cases (a) and (b), the minimizer of $L_2$, $r^*(x)$, equals the Bayes Classifier of rejector $r^B(x)$.

In summary, we prove that $(r^B(x), e^B(x)) = (r^*(x), e^*(x))$ for any $x$, so the stage-switching surrogate loss function is consistent with the generalized 0-1 loss function.

# F  STOCHASTIC POST-HOC ALGORITHM FOR BOUNDED REJECT RATE SETTING

We propose a stochastic post-hoc method, which is shown in Algo. 3, Algo. 4 and Algo 5. In this algorithm for BRR settings, we firstly train our rejector and server classifier following the process as shown in Algo. 3, which is similar to algorithms for PPR or IA settings (can also be trained with asynchronized algorithm 2). After training, we use a calibration set $D_{\text{cali}}$ to calculate the empirical reject rate $q_1$, based on the output of rejector $r(x)$. Once we get the empirical reject rate, we compare it with the bounded reject rate $q$.

If $q$ is greater than $q_1$, that means we have to sacrifice several rejected samples to force them to make prediction locally at client classifier $m(x)$; the input samples are following the Algo. 4, where each rejected samples are made a final decision by the value of a uniform random variable. Once the value is below the ratio $p$, the sample will still be sent to the server. Otherwise, it may be predicted by the client classifier. When the empirical reject rate is higher than the bounded reject rate, after adding this post-hoc mechanism, the bounded reject rate $q$ can just ensured.

If $q$ is smaller than or equal to $q_1$, that means we can reject more times than the rejector $r(x)$ request. Then, we follow a similar mechanism in Algo. 5, which uses a random variable to decide which samples the rejector sends to the server classifier. After using the post-hoc mechanism, we can ensure the bounded reject rate while making the best use of the server classifier.

---

**Algorithm 3** Training Process of Stochastic Post-hoc Method

---

**Input:** Training set $D = \{(x_i, y_i) : i \in [n]\}$, Calibration set $D_{\text{cali}} = \{(x_i, y_i) : i \in [k]\}$, Fixed client classifier $m$, Rejector $r^0$, Sever classifier $e^0$, Bounded reject rate $q$.

1: **for** $t = 1$ to $n$ **do**
2:     $L_1^t(x_t, y_t) = -\ln \frac{\exp(e_{y_t}(x_t))}{\sum_j \exp(e_j(x_t))}$
3:     $e^t \leftarrow \text{SGD}(L_1^t(x_t, y_t), e^{t-1})$
4:     $L_2^t(x_t, y_t) = -(1 - c_e - c_1 + c_1 \mathbf{1}_{e^t(x_t)=y_t}) \ln \frac{\exp(r_2(x_t))}{\exp(r_2(x_t)) + \exp(r_1(x_t))} - \mathbf{1}_{m(x_t)=y_t} \ln \frac{\exp(r_1(x_t))}{\exp(r_2(x_t)) + \exp(r_1(x_t))}$
5:     $r^t \leftarrow \text{SGD}(L_2(x_t, y_t), r_{t-1})$
6: **end for**
7: $q_1 \leftarrow \text{EmpiricalRejectRate}(D_{\text{cali}})$
8: **return** $r^n, e^n, q, q_1$

---

**Algorithm 4** Inference Phase of Stochastic Post-hoc Method When $q < q_1$

---

**Input:** Client Classifier $m$, Trained Rejector $r^n$, Trained Sever Classifier $e^n$, Input Sample $x$, Bounded reject rate $q$, empirical reject rate $q_1$.

1: $p = q/q_1$
2: **if** $r(x) \leq 0$ **then**
3:     Sample $i$ from $(0, 1)$ uniform distribution.
4:     **if** $i \leq p$ **then**
5:         $\hat{y} \leftarrow e^n(x)$
6:     **else**
7:         $\hat{y} \leftarrow m(x)$
8:     **end if**
9: **else**
10:     $\hat{y} \leftarrow m(x)$
11: **end if**
12: **return** $\hat{y}$

---

---

**Algorithm 5** Inference Phase of Stochastic Post-hoc Method when $q \geq q_1$

---

**Input:** Client Classifier $m$, Trained Rejector $r^n$, Trained Sever Classifier $e^n$, Input Sample $x$,
  Bounded reject rate $q$, empirical reject rate $q_1$.
1:   $p = (q - q_1)/(1 - q_1)$
2: **if** $r(x) > 0$ **then**
3:      Sample $i$ from $(0, 1)$ uniform distribution.
4:      **if** $i \leq p$ **then**
5:          $\hat{y} \leftarrow e^n(x)$
6:      **else**
7:          $\hat{y} \leftarrow m(x)$
8:      **end if**
9: **else**
10:     $\hat{y} \leftarrow e^n(x)$
11: **end if**
12: **return** $\hat{y}$

---

## G   EXPERIMENT DESCRIPTION AND ADDITIONAL RESULTS

**Datasets**   We test our proposed surrogate loss function 7 and algorithms for different settings on CIFAR-10 (Krizhevsky & Hinton, 2009), SVHN (Netzer et al., 2011) and CIFAR-100 (Krizhevsky & Hinton, 2009) data sets. CIFAR-10 consists of $32 \times 32$ color images drawn from 10 classes and is split into 50000 training and 10000 testing images. CIFAR-100 has 100 classes containing 600 images each. SVHN is obtained from house numbers with over 600000 photos in Google Street View images and has 10 classes with $32 \times 32$ images centered around a single character. The experiments are conducted in `RTX 3090`. It takes approximately 5 minutes to train the local model and 30 minutes to train each remote server classifier with the rejector when server classifier is set to be AlexNet or 5 hours to train each remote server classifier with the rejector when server classifier is set to be ViT.

**Comparison over synchronization and asynchronization**   We conduct additional experiments training the server model and rejector under both synchronized, as shown in Algo 1 and asynchronized settings as shown in Algo 2. The model is evaluated in the same way as in Section 5. The contrastive evaluation and loss curve are shown in Table 3 and Figure 4.

**Result for different $c_1$ and $c_e$ on CIFAR-10 and SVHN**   Besieds the result we show in Section 5, we add extensive result for contrastive evaluation with different inaccuracy cost $c_1$, reject cost $c_e$ and synchronization interval $S$ (for IA setting) for CIFAR-10 and SVHN with local classifier and rejector being LeNets and remote classifier being AlexNet. The results are shown in Table 3.

**Result for different $c_1$ and $c_e$ on CIFAR-100**   Since our stage-swtich surrogate loss function doesn't set up constraints on the size of dataset and structures of machine learning methods, we test the Contrastive Evaluation on dataset CIFAR100 with local classifier and rejector being LeNet and remote classifier being ViT, a transformer-based vision neural network. The results are shown in Table 4. Both higher $c_1$ and $c_e$ can reduce the ratio of numbers that are sent to remote classifier, but in our experimental results, the reject rate is more sensitive to $c_e$. When $c_e$ is close to 1, almost no sample is sent to remote classifier, which make sense because $c_e = 1$ means that the cost of asking for help is always greater than locally prediction.

**Samples partitioned by rejector**   We choose one case where the rejector and remote classifier AlexNet are trained on SVHN when $c_1 = 1.25$ and $c_e = 0.25$. Then we let rejector partition the test set and randomly pick images from the subset with $r(x) = \text{REMOTE}$ and $r(x) = \text{LOCAL}$, respectively. The results are shown in Fig. 5 and Fig. 6. It's obvious that samples kept locally are clearer, contrast, focused, with high-resolution, while the samples that are supposed to send to the server are blurry, unfocused, with low-resolution. Without any manual adjustment, our rejector learns to assess "difficulty" using the same metrics as humans, by interacting with both local model and server model during the training process.

Table 3: Contrastive Evaluation Results for different cost and synchronization interval

| $S$ | $c_1$ | $c_e$ | | cifar10 (%) | | | SVHN (%) | | |
|---|---|---|---|---|---|---|---|---|---|
| | | | | m | e | differ. | m | e | differ. |
| sync. | 1.0 | 0.25 | $r(x) = \text{LOCAL}$ | 74.8 | 82.8 | **8.0** | 90.1 | 93.2 | **3.1** |
| | | | $r(x) = \text{REMOTE}$ | 54.5 | 68.6 | **14.1** | 62.0 | 72.5 | **10.5** |
| sync. | 1.25 | 0.25 | $r(x) = \text{LOCAL}$ | 73.9 | 81.9 | **8.0** | 90.6 | 93.3 | **2.7** |
| | | | $r(x) = \text{REMOTE}$ | 54.5 | 67.7 | **13.2** | 61.2 | 72.8 | **11.6** |
| sync. | 1.25 | 0.0 | $r(x) = \text{LOCAL}$ | 82.3 | 86.8 | **4.6** | 97.2 | 97.9 | **0.7** |
| | | | $r(x) = \text{REMOTE}$ | 62.6 | 73.8 | **11.2** | 87.6 | 90.8 | **3.2** |
| 100 | 1.0 | 0.25 | $r(x) = \text{LOCAL}$ | 75.9 | 84.3 | **8.4** | 90.6 | 92.9 | **2.3** |
| | | | $r(x) = \text{REMOTE}$ | 55.3 | 69.5 | **14.1** | 63.6 | 72.7 | **9.1** |
| 100 | 1.25 | 0.25 | $r(x) = \text{LOCAL}$ | 74.5 | 83.3 | **8.8** | 91.4 | 93.5 | **2.2** |
| | | | $r(x) = \text{REMOTE}$ | 55.5 | 69.9 | **14.4** | 65.4 | 73.9 | **8.5** |
| 100 | 1.25 | 0.0 | $r(x) = \text{LOCAL}$ | 81.4 | 87.4 | **6.0** | 97.0 | 97.8 | **0.8** |
| | | | $r(x) = \text{REMOTE}$ | 62.6 | 75.3 | **12.7** | 87.3 | 90.6 | **3.3** |
| 1000 | 1.0 | 0.25 | $r(x) = \text{LOCAL}$ | 74.8 | 83.1 | **8.2** | 90.5 | 93.2 | **2.7** |
| | | | $r(x) = \text{REMOTE}$ | 54.7 | 69.2 | **14.6** | 64.3 | 72.9 | **8.6** |
| 1000 | 1.25 | 0.25 | $r(x) = \text{LOCAL}$ | 75.2 | 83.3 | **8.1** | 90.8 | 93.9 | **3.1** |
| | | | $r(x) = \text{REMOTE}$ | 56.0 | 69.7 | **13.7** | 65.1 | 75.6 | **10.5** |
| 1000 | 1.25 | 0.0 | $r(x) = \text{LOCAL}$ | 80.3 | 87.7 | **7.4** | 97.3 | 98.0 | **0.8** |
| | | | $r(x) = \text{REMOTE}$ | 62.8 | 74.5 | **11.7** | 86.9 | 90.7 | **3.7** |
| $|D|$ | 1.0 | 0.25 | $r(x) = \text{LOCAL}$ | 74.1 | 82.9 | **8.9** | 90.1 | 93.0 | **2.9** |
| | | | $r(x) = \text{REMOTE}$ | 56.8 | 71.1 | **14.3** | 62.9 | 73.3 | **10.4** |
| $|D|$ | 1.25 | 0.25 | $r(x) = \text{LOCAL}$ | 72.8 | 82.9 | **10.1** | 90.8 | 93.7 | **2.9** |
| | | | $r(x) = \text{REMOTE}$ | 57.1 | 70.4 | **13.3** | 63.8 | 74.1 | **10.3** |
| $|D|$ | 1.25 | 0.0 | $r(x) = \text{LOCAL}$ | 78.9 | 86.8 | **7.9** | 96.2 | 97.5 | **1.3** |
| | | | $r(x) = \text{REMOTE}$ | 62.8 | 74.9 | **12.0** | 87.5 | 90.9 | **3.3** |

Table 4: Contrastive Evaluation Results for different costs on CIFAR-100 with remote classifier being ViT

| $S$ | $c_1$ | $c_e$ | | cifar100 (%) | | | | |
|---|---|---|---|---|---|---|---|---|
| | | | | ratio | m | e | differ. | joint accuracy |
| sync. | 1.0 | 0.2 | $r(x) = $ LOCAL | 0.1 | 78.6 | 100.0 | **21.4** | 88.1 |
| | | | $r(x) = $ REMOTE | 99.9 | 27.5 | 88.1 | **60.6** | |
| sync. | 1.0 | 0.4 | $r(x) = $ LOCAL | 1.2 | 69.8 | 94.8 | **25.0** | 88.1 |
| | | | $r(x) = $ REMOTE | 98.8 | 27.1 | 88.3 | **61.2** | |
| sync. | 1.0 | 0.6 | $r(x) = $ LOCAL | 13.3 | 46.9 | 90.9 | **43.9** | 82.4 |
| | | | $r(x) = $ REMOTE | 86.7 | 24.6 | 87.9 | **63.3** | |
| sync. | 1.0 | 0.8 | $r(x) = $ LOCAL | 76.4 | 30.7 | 88.8 | **58.0** | 43.4 |
| | | | $r(x) = $ REMOTE | 23.6 | 17.2 | 84.7 | **67.5** | |
| sync. | 1.0 | 0.95 | $r(x) = $ LOCAL | 100.0 | 27.6 | 88.2 | **60.7** | 27.6 |
| | | | $r(x) = $ REMOTE | 0.0 | N/A | N/A | N/A | |
| sync. | 1.9 | 0.5 | $r(x) = $ LOCAL | 4.2 | 58.6 | 92.7 | **34.1** | 86.8 |
| | | | $r(x) = $ REMOTE | 95.8 | 26.2 | 88.0 | **61.9** | |
| sync. | 1.9 | 0.75 | $r(x) = $ LOCAL | 50.8 | 35.7 | 89.5 | **53.8** | 60.7 |
| | | | $r(x) = $ REMOTE | 49.2 | 19.1 | 86.6 | **67.5** | |
| sync. | 2.0 | 0.2 | $r(x) = $ LOCAL | 0.1 | 100.0 | 100.0 | **0.0** | 88.1 |
| | | | $r(x) = $ REMOTE | 99.9 | 27.5 | 88.1 | **60.6** | |
| sync. | 2.0 | 0.4 | $r(x) = $ LOCAL | 1.1 | 67.3 | 94.4 | **27.1** | 87.6 |
| | | | $r(x) = $ REMOTE | 98.9 | 27.1 | 87.8 | **60.7** | |
| sync. | 2.0 | 0.6 | $r(x) = $ LOCAL | 13.5 | 47.2 | 90.9 | **43.7** | 82.1 |
| | | | $r(x) = $ REMOTE | 86.5 | 24.5 | 87.6 | **63.1** | |
| sync. | 2.0 | 0.8 | $r(x) = $ LOCAL | 76.4 | 30.8 | 89.1 | **58.3** | 43.6 |
| | | | $r(x) = $ REMOTE | 23.6 | 17.0 | 85.2 | **68.2** | |
| sync. | 2.0 | 0.95 | $r(x) = $ LOCAL | 100.0 | 27.6 | 88.0 | **60.4** | 27.6 |
| | | | $r(x) = $ REMOTE | 0.0 | N/A | N/A | N/A | |
| sync. | 3.0 | 0.2 | $r(x) = $ LOCAL | 0.1 | 100.0 | 100.0 | **0.0** | 87.8 |
| | | | $r(x) = $ REMOTE | 99.9 | 27.5 | 87.8 | **60.3** | |
| sync. | 3.0 | 0.4 | $r(x) = $ LOCAL | 1.1 | 66.1 | 93.9 | **27.8** | 87.8 |
| | | | $r(x) = $ REMOTE | 98.9 | 27.1 | 88.1 | **61.0** | |
| sync. | 3.0 | 0.6 | $r(x) = $ LOCAL | 13.9 | 46.9 | 90.1 | **43.3** | 82.2 |
| | | | $r(x) = $ REMOTE | 86.1 | 24.4 | 87.9 | **63.5** | |
| sync. | 3.0 | 0.8 | $r(x) = $ LOCAL | 78.5 | 30.5 | 88.8 | **58.3** | 42.3 |
| | | | $r(x) = $ REMOTE | 21.5 | 16.7 | 85.4 | **68.7** | |
| sync. | 3.0 | 0.95 | $r(x) = $ LOCAL | 100.0 | 27.6 | 88.0 | **60.4** | 27.6 |
| | | | $r(x) = $ REMOTE | 0.0 | N/A | N/A | N/A | |
| sync. | 4.0 | 0.2 | $r(x) = $ LOCAL | 0.1 | 100.0 | 100.0 | **0.0** | 60.4 |
| | | | $r(x) = $ REMOTE | 99.9 | 27.5 | 87.9 | **60.4** | |
| sync. | 4.0 | 0.4 | $r(x) = $ LOCAL | 1.1 | 65.7 | 95.2 | **29.5** | 87.8 |
| | | | $r(x) = $ REMOTE | 99.0 | 27.1 | 88.0 | **60.8** | |
| sync. | 4.0 | 0.6 | $r(x) = $ LOCAL | 13.6 | 46.4 | 90.5 | **44.1** | 81.9 |
| | | | $r(x) = $ REMOTE | 86.4 | 24.6 | 87.5 | **62.9** | |
| sync. | 4.0 | 0.8 | $r(x) = $ LOCAL | 78.5 | 30.6 | 88.7 | **58.1** | 42.4 |
| | | | $r(x) = $ REMOTE | 21.5 | 16.5 | 85.3 | **68.8** | |
| sync. | 4.0 | 0.95 | $r(x) = $ LOCAL | 100.0 | 27.6 | 87.8 | **60.2** | 27.6 |
| | | | $r(x) = $ REMOTE | 0.0 | N/A | N/A | N/A | |
| sync. | 5.0 | 0.2 | $r(x) = $ LOCAL | 0.1 | 100.0 | 100.0 | **0.0** | 87.9 |
| | | | $r(x) = $ REMOTE | 99.9 | 27.5 | 87.9 | **60.4** | |
| sync. | 5.0 | 0.4 | $r(x) = $ LOCAL | 1.1 | 68.7 | 95.7 | **27.0** | 87.6 |
| | | | $r(x) = $ REMOTE | 98.9 | 27.1 | 87.8 | **60.7** | |
| sync. | 5.0 | 0.5 | $r(x) = $ LOCAL | 4.1 | 57.8 | 93.4 | **35.7** | 86.6 |
| | | | $r(x) = $ REMOTE | 95.9 | 26.3 | 87.8 | **61.6** | |
| sync. | 5.0 | 0.6 | $r(x) = $ LOCAL | 13.6 | 49.4 | 90.6 | **41.1** | 82.5 |
| | | | $r(x) = $ REMOTE | 86.4 | 24.1 | 87.7 | **63.6** | |
| sync. | 5.0 | 0.8 | $r(x) = $ LOCAL | 79.3 | 30.5 | 88.8 | **58.3** | 41.9 |
| | | | $r(x) = $ REMOTE | 20.7 | 16.4 | 85.4 | **69.1** | |
| sync. | 5.0 | 0.95 | $r(x) = $ LOCAL | 100.0 | 27.6 | 87.9 | **60.4** | 27.6 |
| | | | $r(x) = $ REMOTE | 0.0 | N/A | N/A | N/A | |

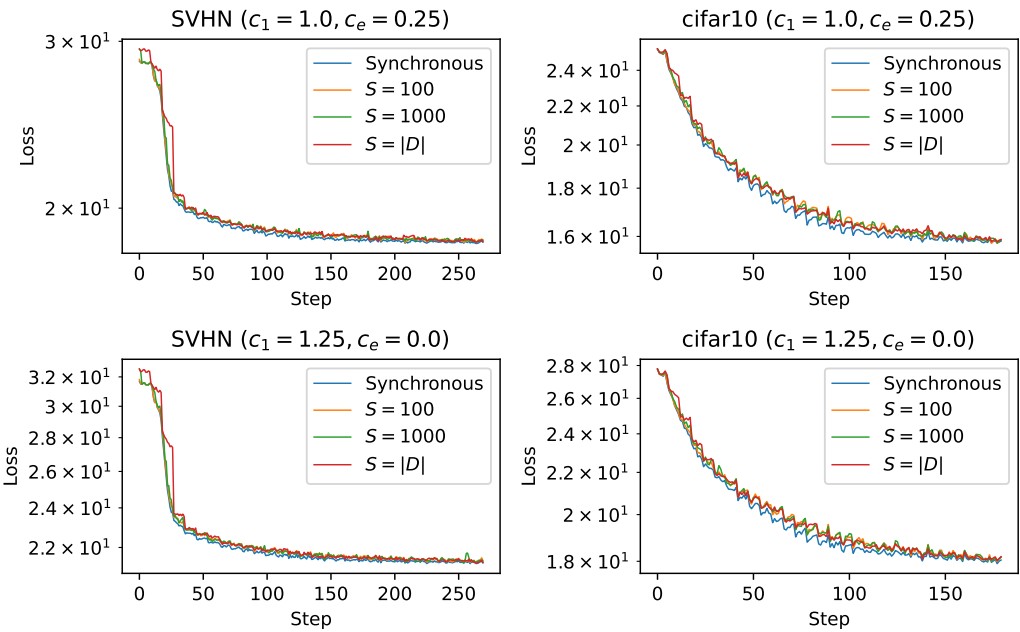

Figure 4: Comparison of synchronization and synchronization with different parameters

Table 5: Accuracy of classifiers on CIFAR-10 when client classifier is pre-trained without "truck" class

|  | "truck" (%) | other classes (%) | all classes (%) |
|---|---|---|---|
| only local classifier | 0 | 64.4 | 58.0 |
| only remote classifier | 87.5 | 72.5 | 74.0 |
| jointly work | 85.2 | 70.5 | 72.0 |
| rejected rate under jointly work | 96.1 | 63.1 | 72.7 |

**Experiments on imbalance dataset**   We set up an additional experiment to evaluate the impact of imbalance dataset issues for our multi-class L2H. We choose one case where the rejector and remote classifier AlexNet are trained on CIFAR-10 or SVHN when $c_1 = 1.25$ and $c_e = 0.25$. In this experiment, we pre-train the local model with the reduced dataset that doesn't contain any sample from one certain class (no "truck" in CIFAR-10 and no "9" in SVHN). Based on that, we train rejector and edge model with a full dataset. The results in Table 5 and Table 6 show that almost all the samples from missing class are identified by rejector and sent to server model. The overall accuracy doesn't get undermined compared to standard setting. The training of the rejector makes the multi-class L2H robust to imbalance dataset or distribution drifting issues. And after training, the remote classifier becomes expert that is specialized on the missing class.

Table 6: Accuracy of classifiers on SVHN when client classifier is pre-trained without "9" class

|  | "9" (%) | other classes (%) | all classes(%) |
|---|---|---|---|
| only local classifier | 0 | 92.2 | 83.0 |
| only remote classifier | 94.5 | 89.5 | 90.0 |
| jointly work | 90.0 | 88.9 | 89.0 |
| rejected rate under jointly work | 93.5 | 12.0 | 17.8 |

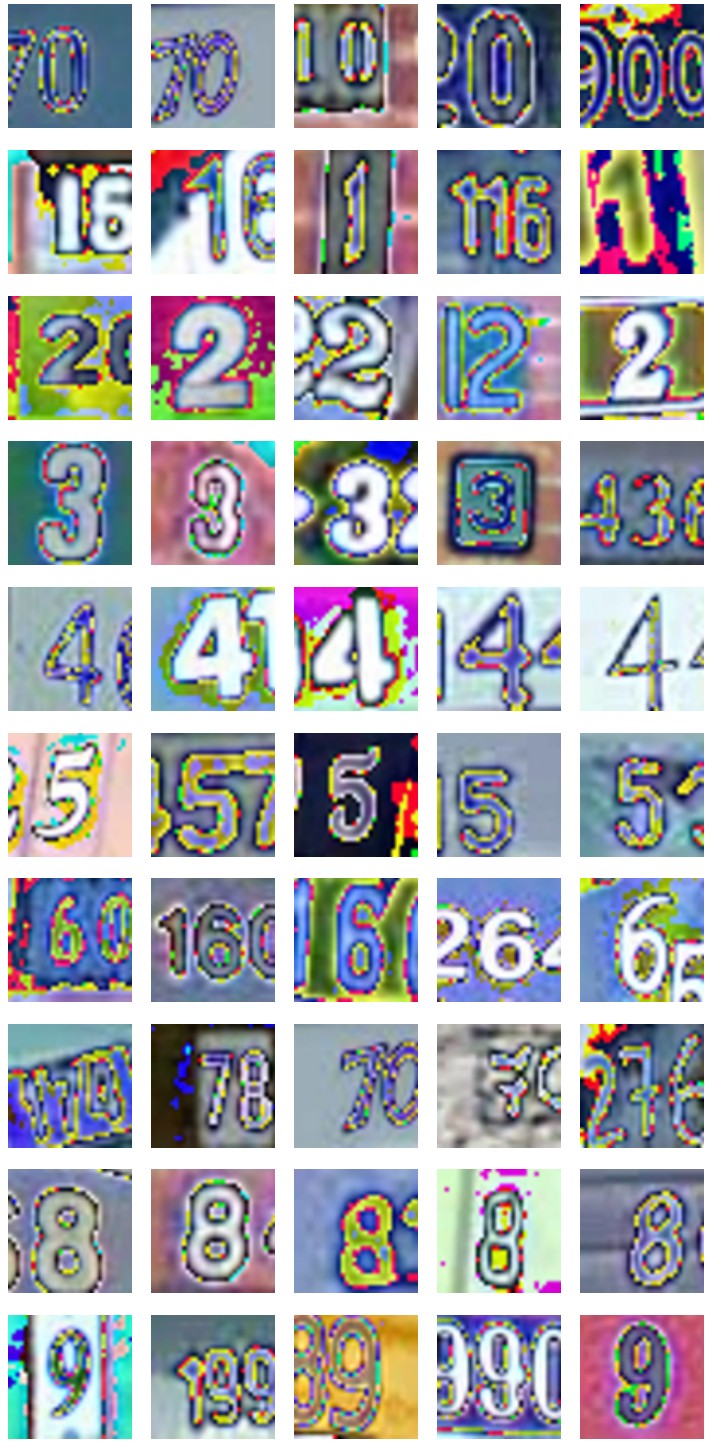

Figure 5: Samples from SVHN with $r(x) = \textsc{local}$

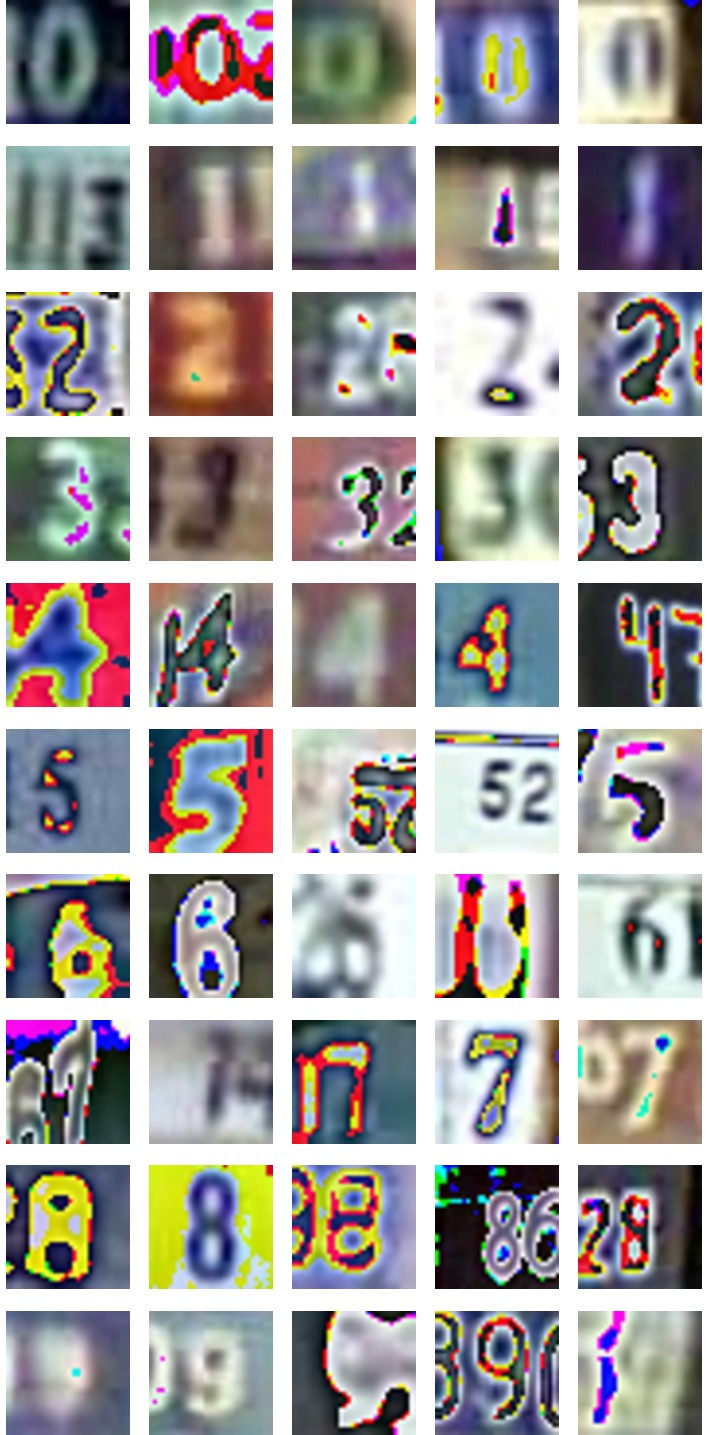

Figure 6: Samples from SVHN with $r(x) = \textsc{remote}$

Table 7: Contrastive Evaluation on Training Asynchronously with $S = 100$, $c_1 = 1.25$ and $c_e = 0.25$

|  | cifar10 (%) | | | SVHN (%) | | |
|---|---|---|---|---|---|---|
|  | m | e | differ. | m | e | differ. |
| data with rejector $r(x) = \text{LOCAL}$ | 74.5 | 83.3 | **8.8** | 91.4 | 93.5 | **2.2** |
| data with rejector $r(x) = \text{REMOTE}$ | 55.5 | 69.9 | **14.4** | 65.4 | 73.9 | **8.5** |

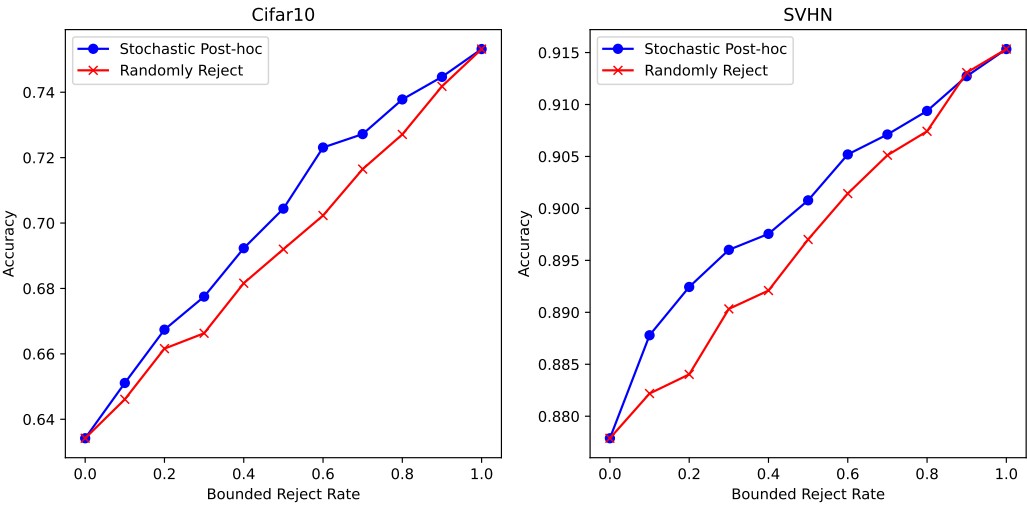

Figure 7: Comparison with randomly reject after Stochastic Post-hoc Algorithm when $c_e = 0.25$ and $c_1 = 1.12$

**Accuracy comparison over stochastic post-hoc and randomly reject on bounded reject rate setting** To show that the client classifier under Bounded Reject Rate settings would benefit from training with Algo. 3 and testing with Algo. 4 and Algo. 5, we set up a randomly reject method as a baseline. In this randomly reject method, we assume that there is no rejector, the sample is randomly sent to server classifier according the value of bounded reject rate, e.g., with bounded reject rate $0.5$, half of the samples will be randomly chosen and sent to server for inference. We train our rejector and server classifier by following Algo. 3. After that, given different bounded reject rate, we use either Algo. 4, or Algo. 5 to strictly control the actual reject rate below the bound. The results of the accuracy-vs-bounded reject rate for the Stochastic Post-hoc method and the random reject method are shown in Fig. 7. The result demonstrates that by using the Stochastic Post-hoc Algorithm, the system can still actively pick up the samples that fit the server classifier even under the constraints of a bounded reject rate.

