# OpenReview forum: "Learning to Help in Multi-Class Settings"
_ICLR.cc/2025/Conference — ICLR 2025 Poster_

### Official Review · Reviewer_vq4j · 2024-11-01

**Soundness:** 2
**Presentation:** 3
**Contribution:** 2
**Rating:** 6
**Confidence:** 4

**Summary:**

This paper extends the L2H model from binary classification to multi-class settings, enabling collaboration between a local (client) model and a remote (server) model, particularly tailored for resource-constrained environments. A stage-switching surrogate loss function is developed to train the server model and a rejector, which facilitates selective sample deferral to the server, thereby optimizing overall performance.

**Strengths:**

+ This paper contributes by extending the L2H framework to multi-class classification, which enhances the applicability of client-server cooperation in environments with limited resources.

+ The surrogate loss function supports asynchronous training, which aligns with scenarios of intermittent server access, beneficial for distributed learning settings.

+ This paper provides a solid theoretical foundation for the proposed method, supporting its validity and enhancing confidence in its application.

**Weaknesses:**

- The client-server setup involves data offloading, raising potential privacy risks. The paper lacks a discussion on privacy-preserving methods, such as differential privacy, which are common in federated learning.

- While the surrogate loss aims to approximate the Bayes optimal classifier, the paper does not provide rigorous proofs for its consistency or convergence in multi-class settings, particularly under non-IID data distributions, a common issue in federated learning.

- The approach assumes IID data, which may not reflect real-world federated learning scenarios. Non-IID data could impair the rejector’s effectiveness in making deferral decisions, limiting the model's applicability.

- While the asynchronous training algorithm partially addresses intermittent server availability, it does not fully address the issues of model drift or maintain accuracy under unstable connectivity. Adaptive adjustments for synchronization frequency and performance are not fully explored.

- Relying on a fixed client classifier limits its ability to adapt to new data distributions or improve through collaborative training, potentially degrading performance over time and increasing dependency on the server model.

- The experimental setup involves relatively simple architectures (LeNet-5 for the client and AlexNet for the server). This limits the generalization of the framework to applications involving more complex models, such as those used in large-scale real-world systems. Additionally, the conclusion’s mention of "multi-party collaboration in hybrid machine learning systems, especially in the era of LLMs." could be misleading, as the paper does not provide any experiments or detailed discussion about large language models.

**Questions:**

- Could the authors consider privacy-preserving techniques, such as differential privacy, to mitigate potential privacy risks? Additionally, how would these methods impact the proposed approach's effectiveness?

- Can the paper provide theoretical justifications for the surrogate loss's consistency and convergence properties, especially in non-IID data conditions?

- How could the framework adapt to handle non-IID data distributions across clients? Could the rejector's decision-making process be adjusted for such scenarios?

- For asynchronous training, could a method be introduced to adaptively adjust synchronization interval $S$, improving applicability to real-world deployments?

- Is there a way to make the client classifier adaptable to changing data distributions over time?

- Would the method be effective with more complex models, such as GPT-2, as hinted in the conclusion?

---

> ### Author Response · Authors · 2024-11-21
> **Responses to Weakness:**
>
> We appreciate the reviewer’s insightful observations and constructive criticism. The reviewer raises several questions about assumptions related to federated learning. In particular, there are many questions about non-IID data (we assume across multiple clients) and differential privacy for offloading. Again, we emphasize that we are not examining a federated learning problem in this work.
>
> We respond to the reviewer’s comments by translating their overall concern to our problem setting.
>
> **Responses to Weakness:**
>
> #### 1. **\[lack of discussion on privacy issues\]**
>
> We can certainly add some discussion in future work on privacy issues in our L2H setup.
>
>    Privacy concerns may arise in the training phase, which could be addressed by training using differentially private SGD (DP-SGD) or another standard private optimization approach.
>
>    In deployment/testing, the primary privacy concern would be if the server is not trusted. Because each client is an individual, they must use a local differentially private mechanism during offloading. This, in turn, would degrade the signal, making the server’s performance less helpful. Indeed, depending on the privacy parameters, there may be no benefit in offloading. Rather than differential privacy, as the reviewer suggests, some solutions based on trusted execution environments (TEEs) may be more appropriate. However, these come with their own issues in terms of security guarantees.
>
> #### 2. **\[proof of consistency\]**
>
> The proof of consistency requested by the reviewer is in Appendix E.
>
> #### 3. **\[non-IID distributions\]**
>
> The main contribution of our paper is in the theory/concept, so the natural starting point is to make the simplest assumptions to allow for a tractable analysis. Under explicit models for dependence or specific application scenarios, we may be able to extend the analysis, but it is unclear how much insight this would give.
>
>    A related issue, again in deployment, is that the samples presented to the rejector will be correlated across time. For example, in screening video frames, there will be significant dependence across frames. We believe that this is application-specific: if we expect temporal dependence, we can incorporate this into the rejector by potentially adding memory to the decision process. This is an interesting direction for future work and seems non-trivial to formulate.
>
>    Since we are focusing on a single client and not a federated learning setup, the more relevant issue for us is not non-i.i.d. data across clients but instead extreme label imbalance within our given client. We add one experiment for the evaluation of the imbalance training set for the pre-trained client model in Table 5 and Table 6 of Appendix G. The result shows that even though the training set has changed, the joint training of the rejector and remote model will adapt to this distribution change and keep the same level of accuracy. For example, in Table 6, the overall accuracy for the local model is only **83%** since this classifier wasn’t trained on class “9”. If we only test the remote model on the whole dataset, the overall accuracy is around **90%**. Through our joint training process, the rejector is specialized to identify the “unseen” 9s with a reject rate (ratio of samples sent to remote) **93.5%** for this class, while the overall reject rate for all classes is only **17.8%**. However, with a relatively low reject rate, the overall accuracy of our L2H system is around **89.0%**, which is quite close to the accuracy of only using the remote classifier. That means our L2H method can selectively send samples to a remote model for help, effectively balancing the accuracy and latency. Our surrogate loss function is robust in dealing with imbalanced dataset issues.
>
> (continued in next comment)

---

> ### Author Response · Authors · 2024-11-21
> **Responses to Weakness (continued):**
>
> #### 4. **\[model drift issue\]**
>
> Intermittent availability (IA) setting is one of three scenarios that we are interested in. We appreciate reviewer vq4j pointed out the issue of model drift. This issue does reduce the rate of convergence during the training phase as shown in Figure 3 and Figure 4\. The convergence rate difference between asynchronous training and synchronous training depends on the structure of loss functions, initializations and specific optimization techniques that have been chosen. We analyzed the case when the loss function is strongly convex and Lipshicz. Referring to our response to Question 2 from reviewer FjCZ, we show that with the model drift issue, asynchronous algorithm converges slower than the synchronous algorithm with factor of $( 1 \- \\frac{m}{N\_e} )^{1-S}$. On the other hand, with different synchronization frequencies, the surrogate loss functions are converging to the same level. This result coincides with our Theorem 3.2 (Consistency of surrogate loss) that the minimizer of our surrogate loss function doesn’t depend on the optimization algorithms no matter if it is synchronized or unsynchronized. We will take the adaptive synchronization frequency as our future direction with more advanced scenarios for IA settings.
>
> #### 5. **\[degrading performance over time and increasing dependency on the server model\]**
>
> a. We didn’t keep the client classifier fixed on purpose. The setting is from the situations in real world scenarios. There are tons of light-weight machine learning models with physical deployments such as “smart infrastructure” or automotive applications. They are pre-trained offline and then deployed to edge devices like embedded systems, car chips or phones. Replacing them with new devices is expensive and requires massive labor. On the other hand, offloading all the tasks to a remote server might be another solution but will cause latency issues. In that sense, we consider a L2H framework to train a rejector and remote server while letting the rejector actively choose the partial samples for offloading. L2H helps balance the tradeoff between accuracy and latency since the server classifier is on the remote side with connection delay.
>
> b. With the collaboration of rejector and server classifier, we can extend the usability of our overall system while not relying too much on server classifier. As we mentioned in previous response on \[non-IID distributions\], the experiment on imbalance dataset shows that under the change of distribution, the rejector can still make the best use of the client classifier. With only 17.8% rejected data samples, the overall accuracy is almost the same level as the server classifier.
>
>
>
> #### 6. **\[model size\]**
>
> Since our theoretical analysis doesn’t depend on the size of machine learning models or the dataset, the experiments can be conducted on any type of classification models. We add additional experiments based on the larger dataset CIFAR-10 and larger model ViT. Please refer to our Appendix G for more details on the experiments.
>
> (continued in next comment)

---

> ### Author Response · Authors · 2024-11-22
> **Responses to Questions:**
>
> **Responses to Questions:**
> #### 1. **\[privacy\]**
>
> As mentioned in response to the weaknesses, differential privacy could be used in the training phase using standard approaches such as DP-SGD: the problem does not provide avenues for more sophisticated differentially private techniques. At test time, if we force the client to use local DP, then offloading may no longer be appealing since the server model would have to deal with (very) noisy inputs.
> #### 2. **\[Consistency in non-IID condition\]**
>
> Our theorem and proof for the consistency of our surrogate loss function are in Theorem 3.2 and Appendix E. If we assume the data are non-IID, then the Bayes classifier itself would change: this means that any consistency result we would want to prove would also be tied to the particular nature of non-IID-ness. However, the current analysis on non-IID can’t be directly introduced to our model:
>    a. For non-i.i.d. time series data, regret analysis can be applied under several assumptions (See Section 1.2. in (Shai et al.,2011)), but we do not consider time series data in this paper.
>    b. There is another common type of non-IID data, widely used in prior work (e.g., (McMahan et al., 2017; Tang et al., 2018; Zhao et al., 2018)): skewed distribution of data labels across devices/locations. This type of non-IID data does not work for our system since we assume the client classifier $m(x)$ is fixed. During the training of the rejector and server classifier, the $m(x)$ works as a fixed function independent of the server dataset. That is, the training process reduces to an IID case with only the server dataset.
>
>    For more general types of non-IID data, we need another specific formulation of the problem. Then, we should follow a different set of analyses (instead of consistency, regret bound is more discussed in non-IID papers), together with algorithms to mitigate the effect of non-IID, which is beyond the scope of our discussion in this paper. We will add the discussion on non-IID to our future work.
>
> #### 3. **\[adaptation to non-IID\]**
>
> We are currently considering one client and one server case. If we extend it to multiple clients and one server case, each client would have its own rejector, so if the data at each client is different then the decision rules derived from rejectors themselves will be different.
> #### 4. **\[adaptively synchronization interval\]**
>
> There are many variations of the training algorithms that may be useful in specific applications. The reviewer suggests adapting the number of local iterations is a good idea; we can also adapt batch sizes and other time scales depending on real latency requirements or bandwidth constraints.
> #### 5. **\[change of distribution\]**
>
> In our framework the client classifiers are fixed – this is the constraint in the L2H framework, which differs from the LWA or L2D framework. The motivating applications are scenarios where the client may be running on older hardware which is difficult or expensive to replace. For example, there may be clients at every traffic intersection in a large city; replacing all of these would be extremely expensive. We want to use the rejector to help extend the life of these systems, which would be very helpful if the distributions shifted over time. Indeed, we could retrain the rejector periodically to help in such adaptation.
> #### 6. **\[larger model\]**
>
> Since we are considering classification tasks, instead of GPT-2, we use ViT as the more complex model. As a comparison, GPT-2 contains 124M parameters while ViT contains 632M parameters. ViT is also a transformer-based neural network and SOTA model for classification. Please see our additional experiment in Appendix G.
>
>
> References
>
> \[1\] Shai Shalev-Shwartz. Online Learning and Online Convex Optimization. Foundations and Trends in Machine Learning, 4(2):107–194, 2011\.
>
> \[2\] McMahan, Brendan, et al. "Communication-efficient learning of deep networks from decentralized data." *Artificial intelligence and statistics*. PMLR, 2017\.
>
> \[3\] Tang, Hanlin, et al. "$ D^ 2$: Decentralized training over decentralized data." *International Conference on Machine Learning*. PMLR, 2018\.
>
> \[4\] Zhao, Yue, et al. "Federated learning with non-iid data." *arXiv preprint arXiv:1806.00582* (2018).

---

> > ### Comment · Reviewer_vq4j · 2024-11-26
> >
> > Thank you for making significant improvements and addressing several concerns. The revised version includes thorough proofs of surrogate loss consistency in multi-class settings. Additionally, experiments now cover asynchronous training and more complex models like ViT. These enhancements strengthen the theoretical foundation and improve practical relevance for constrained deployment scenarios.
> >
> > However, some key issues remain unresolved, such as privacy risks in the client-server setup are still underexplored. Although techniques such as differential privacy and trusted execution environments are mentioned, there is no implementation or analysis of their impact. This leaves a critical aspect of hybrid systems inadequately addressed.
> >
> > Nevertheless, I will increase my score. Thank you again for your efforts to improve the work.

---

> ### Author Response · Authors · 2024-11-27
>
> Thank you for your positive feedbacks and helping us in refining our paper. We are also grateful for your kind acknowledgement of the improvements! We will take your suggestions on privacy into consideration in our extended work.

---

### Official Review · Reviewer_FjCZ · 2024-11-02

**Soundness:** 3
**Presentation:** 3
**Contribution:** 3
**Rating:** 8
**Confidence:** 2

**Summary:**

This paper focuses on the collaboration between the client model and the server model by using a rejector. To train the rejector, client model, and server model, this paper derives a stage-switching surrogate loss function to allow the asynchronous training between the client model and the server model.

**Strengths:**

1. The theoretical analysis of this paper is solid, which extends the binary classification setting of L2H to general multi-class settings.

2. The problem is important and the proposed method uses a surrogate loss to asynchronously train client and server model, which is simple yet theoretically effective.

**Weaknesses:**

The experiments are limited. Experiments are mainly composed of hyper-parameter sensitive analysis evaluations, which lack comparison to baselines. For example, this paper should compare the proposed method with those using the synchronous training method or different surrogate loss functions in terms of overall performance.

**Questions:**

1. What's the process during the inference stage with the surrogate loss?

2. What's the convergence rate difference between the asynchronous and synchronous methods?

---

> ### Author Response · Authors · 2024-11-21
> **Responses to Weakness:**
>
> We appreciate the reviewer’s insightful observations and constructive criticism.
>
> **Responses to Weakness:**
>
> 1. **\[experiments\]**
>    This paper focuses more on the theoretical part. The analysis is general for all multi-class classification models. Meanwhile, we also added more experiments in the appendix:
>
>     a. We test our multi-class L2H on a larger dataset CIFAR10 and larger machine learning models ViT, shown in Table 4\.
>
>     b. We test our multi-class L2H on an imbalance dataset, where the data distribution for the pre-trained client model is different from the data distribution for the training of the rejector and server model, shown in Table 5 and Table 6\.
>
>     c. We showcase what kinds of samples are kept locally and what is sent to a remote server, shown in Fig. 5 and Fig. 6\.
>
> 2. **\[lack of comparison\]**
>     a. To our knowledge, this work is the first paper investigating L2H for a multi-class setting. We show in Appendix A that our proposed surrogate loss function is not a straightforward extension from the surrogate loss for binary class by Wu\&Sarwate(2024). The direct extension from their surrogate loss will make the loss function non-differentiable and not trainable by gradient-based optimization algorithms. We also make a detailed comparison with those common surrogate loss functions proposed for Learning with Abstention (LWA) and Learning to Defer (L2D), which are similar but different settings from L2H, in Appendix B and Table 2\. In the LWA framework, they only consider the existence of rejector $r(x)$, and there is no server model $e(x)$. In L2D, there are rejector $r(x)$ and server model $e(x)$, but they assume the server model is a fixed part (like a human expert or fixed model) during training. Through the comparison, we find that previous surrogate loss functions either miss components in L2H, do not address latency and computation concerns, or do not apply to the settings we are interested in. In this sense, we mainly conduct experiments to verify the arguments we make in theoretical analysis.
>
>     b. We compare the convergence rate between sync and async settings in Figure 3 in the main part of the paper and in Figure 4 of Appendix G.
>
>     c. We compare with a threshold-based method for the Bounded Reject Rate(BRR) setting in Appendix G, Figure 7\. Our Stochastic Post-hoc method outperforms the Randomly Reject method for each given reject rate constraint. The result, together with the rejected sample and non-rejected sample showcased in Fig. 5 & 6, show that jointly training rejector $r(x)$ and $e(x)$ with our surrogate loss function could teach the rejector to actively pick out the "hard" samples.
>
> (continued in next comment)

---

> ### Author Response · Authors · 2024-11-21
> **Responses to Questions:**
>
> **Responses to Questions:**
>
> #### 1. **What's the process during the inference stage with the surrogate loss?**
>    Like other classification tasks, the surrogate loss function is only used for the training stage. Because the loss function is used to evaluate our models' performance with current parameters. Our models are updated based on the loss function's gradients (or other quantities). Once the training is finished, the models will be ready to use, so we no longer need to refer to the surrogate loss function. In the inference stage, the pipeline is:
> a. A new sample $x$ is sent to the client side of the L2H system;
> b. The rejector firstly receives the sample and produces output $r(x)$;
> c. If $r(x)=\\text{LOCAL}$, the sample is kept locally and predicted by local model $\\hat{y}=m(x)$; if $r(x)=\\text{REMOTE}$, the sample is sent to server side and predicted by server model $\\hat{y}=e(x)$;
> d. The prediction $\\hat{y}$ is the output of the L2H system for the sample $x$.
>
>
>
> #### 2. **What's the convergence rate difference between the asynchronous and synchronous methods?**
>    The convergence rate depends on the optimization algorithms we used. If we apply the gradient descent method, the analysis is the following:
>    The surrogate loss function consists of $L\_1$ and $L\_2$. In sync and async settings, the updating steps of $L\_1$ are the same, so the convergence rates for this part are also the same. Then, we just need to consider the $L\_2$. The convergence rate depends on the structure of our loss function, initialization points, and the optimization method we use for training. To do the analysis, we add a few assumptions below:
>    Assumption 1: The loss function $L\_2(r, e)$ satisfies the PL-condition (See equation (3) in Karim et al. (2016)) in $r$ and $e$ with constant $m \> 0$.
>    Assumption 2: The gradients of surrogate loss function $\\nabla\_r L\_2 (r, e)$ and $\\nabla\_e L\_2(r, e)$ are Lipschitz with constants $N\_r$ and $N\_e$ respectively.
>    Based on assumptions 1 & 2, if we apply the gradient descent method, the distance between the current loss and the optimal loss is shrinking with factor $(1-m/(N\_r))$ if updates $r$ and with factor $(1-m/(N\_e))$ if update $e$ according to Theorem 1 in Karimi et al. (2016). Starting from $r^{k}$ and $e^{k}$, we consider updating for $S$ times within a communication interval. Then the convergence rate for our synchronous algorithm is $L\_2 (r^{k \+ S}, e^{k \+ 1}) \- L\_2 (r^{\\ast}, e^{\\ast}) \\leq \\left( 1 \- \\frac{m}{N\_r} \\right)^S \\left( 1 \- \\frac{m}{N\_e} \\right) (L\_2 (r^k, e^k) \-L\_2(r^{\\ast}, e^{\\ast}))$ since we update $r$ for $S$ times and update $e$ for once. And the convergence rate for asynchronous algorithm is $L\_2 (r^{k \+ S}, e^{k \+ S}) \- L\_2 (r^{\\ast}, e^{\\ast}) \\leq \\left( 1 \-\\frac{m}{N\_r} \\right)^S \\left( 1 \- \\frac{m}{N\_e} \\right)^S (L\_2 (r^k, e^k) \-L\_2 (r^{\\ast}, e^{\\ast}))$. From the result, the difference is that training with the synchronous algorithm is shrinking faster with extra multiply term $( 1 \- \\frac{m}{N\_e} )^{S-1}$.
>
>
>
> References:
>    \[1\] Karimi, Hamed, Julie Nutini, and Mark Schmidt. "Linear convergence of gradient and proximal-gradient methods under the polyak-łojasiewicz condition." *Machine Learning and Knowledge Discovery in Databases: European Conference, ECML PKDD 2016, Riva del Garda, Italy, September 19-23, 2016, Proceedings, Part I 16*. Springer International Publishing, 2016\.
>    \[2\]Wu, Yu, and Anand Sarwate. "Learning To Help: Training Models to Assist Legacy Devices." *arXiv preprint arXiv:2409.16253* (2024).

---

> > ### Comment · Reviewer_FjCZ · 2024-11-22
> >
> > Thank you for the rebuttal, The responses have addressed most of my concerns.  However, l still think the author should compare their method with more baselines even constructed baselines such as a random uploading strategy or a classification-confidence-based uploading strategy. Overall, the refined paper is a good one and thus I decided to increase my score.

---

> > > ### Author Response · Authors · 2024-11-27
> > >
> > > Thank you for your positive feedbacks. We genuinely appreciate your insightful and constructive comments that greatly enhanced the quality of our work. Thanks again for your suggestions about the comparison experiments. We will consider them in the updated version of our paper.

---

### Official Review · Reviewer_aEUF · 2024-11-08

**Soundness:** 3
**Presentation:** 2
**Contribution:** 2
**Rating:** 6
**Confidence:** 3

**Summary:**

In this manuscript, a hybrid system is introduced, wherein the local model is supplemented by a server-side model and the rejector selects data samples to be directed to either the client model or the server model. The manuscript presents three distinct scenarios: “pay-per-request”, “intermittent availability”, and “bounded reject rate”, which align with practical constraints related to cost, availability, and policy, respectively. To address these challenges, a differentiable and convex stage-switching surrogate loss function is developed to iteratively optimize both the “rejector function” and “server model”. Experimental results validate the superior performance of the proposed framework.

**Strengths:**

- The writing of this manuscript is clear, and easy to follow.
- This manuscript addresses three practical scenarios reflecting constraints related to cost, availability, and policy, and derives formal objective functions to represent them.

**Weaknesses:**

- The authors state that they extend the “learning to help” framework to handle multi-class classification but do not explain the challenges involved in transitioning from binary-class to multi-class classification.
- In the Conclusion section, the authors state that the proposed framework opens new avenues for further exploration in multi-party collaboration. However, since the evaluation uses relatively simple networks, such as LeNet-5 and AlexNet, the authors should include more complex networks to thoroughly validate the framework's performance.
- Figure 2 illustrates that the hyper-parameters $ c_1$ and $c_2$ significantly affect the system's overall performance. Therefore, it is recommended that the authors evaluate the proposed framework's performance while varying $c_1$ and $c_2$ simultaneously. Additionally, the authors should provide guidance on achieving an optimal balance between $c_1$ and $c_2$.
- Table 1 shows that for the CIFAR-10 and SVHN datasets, when the rejector predicts “local”, the performance of $m(\cdot)$ is lower than $e(\cdot)$. Similarly, when the rejector predicts “remote”, $m(\cdot)$ still performs worse than $e(\cdot)$. Typically, the server model’s performance should surpass that of the local model when the rejector predicts “remote”. The authors should clarify this discrepancy.

**Questions:**

As presented in the "Weaknesses" section.

---

> ### Author Response · Authors · 2024-11-21
> **Responses to Weakness:**
>
> We appreciate the reviewer’s insightful observations and constructive criticism.
>
> **Responses to Weakness:**
>
> 1. In Appendix A, we already made a detailed comparison with the surrogate loss function for binary classification proposed in previous work Wu\&Sarwate(2024). Their loss function is in this form: $L\_{\\text{Wu}}(r,e,x,y)=c\_1 \\exp \\left(\\frac{\\beta}{2}(-e y-r)\\right)+c\_e \\exp({-r})+1\_{m(x)y \\le 0}\\exp \\left(\\alpha r\\right)$, where $\\alpha$ and $\\beta$ are parameters for calibration. Here are the reasons why this loss function is not applicable for the transition to multi-class:
>
>    a. In their binary case, they let $e(x)\\leq 0$ and $e(x)\>0$ represent class \-1 and 1 and let multiplication $y e(x)\>0$ and $ y e(x)\<0 $ represent correct and wrong prediction (the same operation for $m(x)$).  $y e(x)$ and $y m(x)$ are key components in their loss function. However, in multi-class classification, the output of $e(x)$ and $m(x)$ are just the index of the predicted class in a discrete space \[K\], where K is the number of classes.  $y e(x)$ and $y m(x)$ can’t be used for measuring the distance between the predicted label and true label in multi-case since they are just the multiplication of two indexes and are not differentiable for training. Our paper defines a new set of notations for multi-class classification. For example, $e(x)$ consists of K sub-function $\[e\_{1}(x), e\_{2}(x), \\cdots, e\_{K}(x)\]$ and the proposed loss function are differentiable for each sub-function.
>
>    b. In their surrogate loss function for binary classification, two calibration parameters  $\\alpha$ and $\\beta$ are required to calculate for each sample. The value of  $\\alpha$ and $\\beta$ depends on the accuracy estimator of local model $m(x)$, which would introduce error into the loss function and the training process.
>
>    c. The first term of the binary loss function is not separable for $e$ and $r$, which means their loss function can’t handle the IA setting where the connection between the local client and the remote server is unstable. The server model and rejector need to be trained asynchronously.
>
> Please refer to Appendix A for a thorough analysis.
>
> 2. We already add experiments on larger datasets with larger server models. The dataset is CIFAR10, which is 10 times larger than the previous dataset in the number of classes, and the model is ViT, which is a transformer-based neural network and the current SOTA framework for classification. The results are attached in Appendix G, and are consistent with the conclusions we draw in our papers.
>
> 3. We add a Table 4 in Appendix G, to simultaneously show the impact of $c\_e$ and $c\_1$ on the performance on the CIFAR100 dataset with server model ViT. Actually, in our setting, $c\_e$ and $c\_1$ are the constraints of our system, and they are not just hyperparameters that can be arbitrarily chosen for training. $c\_e$ is the cost of inquiring about the remote server model. In a real-world system, the inquiry fee, latency, etc. $c\_1$ is paid when the remote server is wrong. In a real-world system, $c\_1$ could be higher than 1 as wrong advice from an expert (like in a medical consultation setting) may cause more serious accidents. But those analyses on Fig. 2, Table 3, and Table 4  are still instructional for designing an L2H system. Referring to equation (6): $r^B \= \\mathbf{1}  \[\\eta\_{j^{\*}(x)} (x) \> (1 \- c\_e \- c\_1) \+ c\_1\\max\_i \\eta\_i (x)\] \\cdot 2 \- 1$, in our paper, with higher $c\_1$ or $c\_e$, the right-hand side inside the indicator function is smaller, which means rejector is more likely to output $\\text{LOCAL}$, that is, more data will be predicted locally. Considering that regression function $\\eta(x)$ is always between 0 and 1 according to the definition, we can derive the feasible interval for $c\_e$ and $c\_1$: $0 \\leq c\_e \<1$ and $0\< c\_1 \< \\frac{1-c\_e}{1- \\max\_{x,i}\\eta\_{i}(x)}$. Once $c_e \geq 1$ and $c_1 \geq \frac{1-c_e}{1- \max_{x,i}\eta_{i}(x)}$, the rejector will keep all samples locally. When $c_e < 0$ and $c_1 \leq 1$, the rejector will send all samples to the server. To design a meaningful L2H system with $c_1$ and $c_e$, the pipeline is to firstly estimate this quantity $\max_{x,i}\eta_{i}(x)$ and find the feasible interval for $c_1$ and $c_e$. Secondly, according to the specific application and demand (e.g., for a system that is latency-sensitive, the $c_e$ might be large, while an accuracy-sensitive system may require smaller $c_e$), choose the $c_1$ and $c_e$ from the feasible intervals.
>
> (continued in next comment)

---

> > ### Comment · Reviewer_aEUF · 2024-11-26
> > **Official Comment by Reviewer aEUF**
> >
> > Thank you for your effort in addressing the concerns raised. While most of my concerns have been addressed, I still believe the proposed system should be tested in more real-world scenarios. The CIFAR-100 dataset is relatively small, with an input size of only 32 × 32 × 3. I recommend evaluating the system on larger datasets, such as ImageNet, DeepFish, or iNaturalist2021. Overall, I have decided to increase my score to 6.

---

> > > ### Author Response · Authors · 2024-11-27
> > >
> > > We deeply appreciate your thoughtful feedback, which has been instrumental in refining our paper. Thank you again for your suggestion on datasets. We will carefully consider them in our revised paper under our resource and computational capability.

---

> ### Author Response · Authors · 2024-11-21
> **Responses to Weakness (continued):**
>
> 4. To clarify, we added one more column to each dataset in Table 1: ratio. The ratio means the percentage of samples from the test set should be sent to “LOCAL” or “REMOTE”. The rejector partitions the test set into two subsets, according to the output of $r(x)$, let’s say $S\_1$ and $S\_2$, where sample $x$ from $S\_1$ always has $r(x)=\\text{LOCAL}$ and sample from $S\_2$ always has $r(x)=\\text{REMOTE}$. In the first row, we test the accuracy of a subset $S\_1$ on both the local model $m(x)$ and server model $e(x)$. In the second row, we test that of subset $S\_2$. In our experiment, the rejector divides the test set into an  “esay” sample set and a “hard” sample set, we attach samples of the set $S\_1$ and $S\_2$ for SVHN in Fig. 5 and Fig. 6\. It shows that samples in $S\_2$ are much blurred, light, and abstract, which explains why both the accuracy of $m(x)$ and $e(x)$ decrease a lot on $S\_2$ compared to those on $S\_1$ (our $e(x)$ is not oracle, it’s just a larger model compared to $m(x)$, that’s why $e(x)$’s performance is also undermined on $S\_2$). But we find that the advantage of $e(x)$ over $m(x)$ is much higher on $S\_2$ than the advantage on $S\_1$ (see column “differ.”), which means let $e(x)$ predict for set $S\_2$ can save more on accuracy. The insight drawn from Table 1 could be that the rejector proactively picks the “hard” samples from the test set and let $m(x)$ only handle “easy” samples and let $e(x)$ only handle “hard” samples.

---

### Official Review · Reviewer_We9X · 2024-11-09

**Soundness:** 3
**Presentation:** 3
**Contribution:** 3
**Rating:** 6
**Confidence:** 2

**Summary:**

This paper extends the ``learning to help'' paradigm from binary classification to multi-class settings. Learning to help aims to learn a rejector for routing between edge legacy devices and server-side large models. This paper provides algorithms with theoretical analysis, followed by experiments on CIFAR-10 and SVHN.

**Strengths:**

The setting of learning to help is quite interesting and meaningful. The authors extend it to multi-class classification cases, which have extensive applications. The proposed method is theoretically guaranteed. Also, the presentation is good.

**Weaknesses:**

The main drawbacks of this paper may be in the experiment part.
- The datasets used are small, such as CIFAR-10 and SVHN. And there are only two datasets. Usually, evaluation on 3 datasets (or more) will be better.
- The used LeNet and AlexNet are tiny. Trending sota models like ViT can be considered.
- The learning to help scenarios can be very useful in large language model settings for on-device inference. Can the authors do some experiments on this? Or at least give a discussion.
- How do you partition the data for training the small legacy model and large server model? Is there any difference?
- I am just wondering about the costs of training the rejector and server model. Can the authors elaborate on this? If the costs are too much compared with the efficiency gains afterwards, why not use the server-side model solely?
- Can the authors do some case studies to show which samples need to be handled on the server instead of on the device? It will help to understand the system.
- Towards ``first principle'' to make the thing simple: for image classification tasks, one possible and simple solution is that during inference, check the prediction confidence (e.g., the softmax entropy) of the device model, and if the confidence is lower than a threshold (a predefined hyperparameter), send it to the server. Is this solution possible? It will not require any additional training for a rejector.

**Questions:**

See weaknesses.

---

> ### Author Response · Authors · 2024-11-21
> **Responses to Weakness:**
>
> We appreciate the reviewer’s insightful observations and constructive criticism.
>
> **Responses to Weakness:**
>
> 1. Because we were focused on the theoretical framework, we provided results for simpler examples as “a proof of concept”. However, the reviewer’s point that our experiments should involve more complex models is well-taken. To address this, we add some experiments on Cifar-100, which contains more samples with 100 classes. The results we concluded from experiments in Cifar-100 are consistent with previous experiments. Please refer to Appendix G for detailed results.
> 2. We accepted reviewers’ suggestions. We add additional experiments on ViT.
> 3. Our tasks executed in the experiment section are image classifications. The closest transformer-based architecture to the language model for image classification is ViT. We test our framework using ViT as the edge net (see Table 4 for more details). Nevertheless, using our framework for the language model, training, and inference is possible. More specifically, as an autoregressive model, the language model uses the previous sequence for the next token prediction and has the same output as the classification problem (probability distribution of the next token). Thus, the proposed framework and the loss function can be directly used for the training process. Then, during the inference time, if we use our framework to generate language in an autoregressive way, some simple language tokens (e.g., prepositions, conjunctions, pronouns) can be generated on local devices very fast, but the language tokens that require reasoning to make a decision can be rejected and sent to remote (e.g., verbs, negating word, nous). This can improve both local device performance and not slow down the generation.
>
> 4. It depends on the scenarios. In the usual case, we consider that the distribution doesn’t change, but the local model is limited in size and computation. In contrast, the remote model is relatively more capable of classification. Then, the two models are trained on the same datasets in different time frames. Considering the imbalance dataset case, we can let local and remote models be trained on different datasets. Please refer to Table 5 and Table 6 for the results. Those different scenarios don’t change our theoretical results. Since we consider that the legacy model is fixed during the rejector and remote model training. The guarantee of consistency of rejector and remote classifier only depends on current distribution $(X, Y)$ and the output of a fixed classifier: $m(x)$ a variable conditioned on $X$, which is independent of $Y$ as shown in the proof of Theorem 3.2. In sum, the learning of the rejector and remote classifier is independent of the distribution previously used for the previous training of the local classifier. In this diagram, if we want to enhance a legacy model to achieve Bayes-optimality, we do not need to recall the (maybe unavailable) training history of that legacy model, which is good news in our interested applications.
>
> 5. In our setting, we assume the component on the local side is lightweight while the model on the remote side is relatively larger. So, jointly training the rejector (lightweight) and remote classifier does add too much computation burden. For example, in our latest experiment, the rejector is LeNet with 60K parameters, and the remote model is AlexNet with 61M or ViT with 632M parameters. With acceptable one-time extra computation on training and reduction of accuracy (for example, with $c\_1=1.25$, $c\_e=0.25$, AlexNet on SVHN, only 1% accuracy reduction compared to remote model solely), each time the rejector chooses to make decision locally, we have those advantages:
>
>    a. For latency-sensitive tasks(like autonomous driving), we will only send samples and consult the remote model when the local model is not confident to predict. The average inference latency is smaller.
>    b. If the remote model is rented from a commercial service and you must pay for each inquiry, the rejector saves money.
>    c. Save computation cost since the local model is smaller.
>
> (continued in next comment)

---

> ### Author Response · Authors · 2024-11-21
> **Responses to Weakness (continued):**
>
> 6. The examples are attached in Fig. 5 and Fig. 6.
> 7. There are two main reasons for not using confidence-threshold methods:
>    a. From a theoretical perspective, the confidence threshold method has its limit. If the reject decision is the confidence score, which is derived from the output of the local model. The reject decision can be written as $r(m(x))$. Then input space becomes a subset of $\\mathcal{X}$ instead of $\\mathcal{X}$ (input space in our rejector); this decision ruler can’t reach Bayes-optimal by optimization in some cases, as shown in Fig.2 of this paper by Cortes et al(2016).
>    b. From an empirical perspective, we noticed a phenomenon in our experiments: the outputs of the softmax layer of our trained classifiers are polarized. The values are concentrated in two small intervals around 0 (say (0, 0.05)) and 1 (say (0.95, 1)) no matter what the true class is. For example, a sample with true class 1 could be predicted as class 2 with a softmax value that is most likely greater than 0.95. The classifier is still too confident in its wrong prediction. This phenomenon is also pointed out in a well-known paper: On Calibration of Modern Neural Networks. That is, deep neural networks are over-confident and poorly calibrated. The output from the softmax layer is not a good estimator of true probability.
>
> Reference
>
> \[1\]Cortes, Corinna, Giulia DeSalvo, and Mehryar Mohri. "Learning with rejection." *Algorithmic Learning Theory: 27th International Conference, ALT 2016, Bari, Italy, October 19-21, 2016, Proceedings 27*. Springer International Publishing, 2016\.

---

> ### Comment · Reviewer_We9X · 2024-11-26
> **Post-rebuttal**
>
> Thanks for the authors' rebuttal. I appreciate the added experiments. I decide to keep the score.

---

> > ### Author Response · Authors · 2024-11-27
> >
> > Thank you for your positive feedbacks! We deeply appreciate the time and effort you’ve dedicated to helping us refine and enhance our work!

---

### Author Response · Authors · 2024-11-21
**Response: common points brought up by reviewers**

We sincerely thank the reviewers for dedicating their time and providing such thoughtful and constructive feedback. Below we address some of the common themes and major points raised in the reviews. More specific responses will appear under the individual reviews. We believe that the points will provide some shared context for all reviewers.

### 1.  **L2H is not a form of federated learning (FL).**

In particular, the target of L2H is to train an extra rejector and the server model to collaboratively work with the existing client model, while the target of FL is to train a single model in a distributed way. The three components local model, rejector, and the server model still engage in the test/deployment phase, while in FL only one model is needed to handle the tasks. More crucially, the dataset is not partitioned in L2H since the client model is fixed during the training phase, while each client model has its own training set on each client site in FL.

While it is also possible to do online training like the form of federated learning, eg. one server vs multiple clients with rejectors, *this would be a topic for future work*. This setting would raise many other practical issues common to all federated learning tasks: non-IID data, privacy, communication complexity, and so on.  As one example, it may be challenging to train a single server model to help multiple heterogeneous clients. As an analogy, it is also difficult for an instructor (=server) to teach (=help) students (=clients) with heterogeneous preparation/background knowledge.

a. The L2H framework is designed to address scenarios in which the *client model cannot be updated once it is deployed*. This can happen if (for example) the client model is hard-coded into hardware. This assumption models many applications involving device-bound models in physical deployments such as “smart infrastructure” or automotive applications.

Once deployed, it is very expensive to physically replace every device: a “safe streets” application may have cameras and computer vision systems at every intersection in a city: replacing all of the devices is like replacing every traffic light.

One proposed architecture for these systems is to offload computation to a cloud service, but this can cause consistent latency for the task.

The L2H model takes advantage of both solutions by letting a rejector only offload partial data to the server while keeping the use of the legacy model. We assume that the client models can be equipped with an updateable rejector so that the rejector and server models can be updated periodically. These updates will happen offline.

b. As with all ML systems, there is a training phase and an inference (deployment) phase for our system. In the training phase, the local model $m(x)$ is a fixed function whose parameters do not change. For each sample $x$, we calculate the loss $L\_1$ and update the parameters of the server model $e(x)$, and then calculate the loss $L\_2$ and update the parameters of reject function $r(x)$.

c. For the inference phase, we do not use the loss function anymore. For each new sample $x$, the client will compute the trained $r(x)$ and decide whether to process $x$ locally with $m(x)$ or send it to the server who will process it with the trained $e(x)$.

d. The standard federated learning (FL) setting is to have multiple clients communicate with a server to train a single model. In this case, a client and server will hold the same model because the server pushes updated models to the clients. In L2H the client model is fixed and the server model is different. The assumption is that the server is more powerful than the client and can therefore have a better model.

Ideally, we would be able to train a simple/lightweight rejector and server to help boost the performance of the overall system without requiring offloading of all test samples.

(continued in next comment)

---

> ### Author Response · Authors · 2024-11-21
> **Response (continued): common points brought up by reviewers**
>
> ### 2. **Why is multi-class L2H (and other related models) challenging compared to the binary case?**
>    a. **Binary L2H:** The direct extension from binary surrogate loss function by Wu & Sarwate (2024) to multi-class would make it non-differentiable and not applicable for gradient-based training. Moreover, it requires estimating parameters for each sample. Finally, the previous model did not consider the three connection scenarios we examine: Intermittent availability (IA) setting. Please refer to Appendix A for more details.
>
>    b. **LWA and L2H:**  In the LWA framework, the rejected sample will be directly discarded without further operation. In L2D, the client can be trained and *the server is a fixed* *model*, like a human expert or pretrained ML model. L2H considers the reverse, where the client is fixed and the server must be trained, so the loss functions from LWA and L2H do not apply directly to this problem since they do not account for training the server. Please refer to Appendix B for more details.
>
>    c. **Confidence-based methods:** Cortes et al. (2016) theoretically prove that confidence-based methods, where the rejector decision depends on the output of the local model, cannot give good solutions for certain types of data distribution. In our preliminary test, the outputs of softmax layers are extremely close to either 0 or 1, which makes it hard to find a practical threshold for rejection. This finding also coincides with the arguments stated by Guo et al. (2017) which imply that  confidence-based methods have intrinsic limitations. Please refer to Appendix B for more details.
>
> ### 3. **Can this multi-class L2H diagram work for large datasets or large models?**
>    a. Our theoretical results characterize the Bayes rule for multi-class L2H, propose a differentiable surrogate loss function, and show that minimizing the surrogate loss is consistent with the Bayes classifiers. Our formulation is general in the sense that it addresses multi-class classification tasks without constraints on the types of dataset and structures of local mode $m(x)$, rejector $r(x)$, and server model $e(x)$. The theory/framework also applies to large models (like ViT) and general datasets and multi-class tasks.
>
>    b. Theory and practice are often different, so as suggested by the reviewers, in the revision we include additional experiments on the larger dataset Cifar100 and the larger model ViT. Cifar-100 contains 100 classes, which is 10 times larger than our previous dataset. ViT is a transformer-based neural network and a SOTA vision classification model with 632M parameters, while our previous LeNet contains 60K parameters and AlexNet contains 61M parameters. The results on these larger models also validate the theoretical analysis. Please refer to Appendix D for more details.
>
> (continued in next comment)

---

> ### Author Response · Authors · 2024-11-21
> **Response (continued): common points brought up by reviewers**
>
> ### 4. **What does the rejector learn from the training process?**
> a. From Table 1 in the main body of our paper and Tables 3 and 4 in Appendix D, we can find that the rejector partitions the dataset samples as \`\`easy’’ or \`\`hard’’ to identify for both local model and server model. In addition, the rejector will send samples to the remote server if the server model has a significant advantage over the local model on those samples: see the column \`\`differ.’’ in each table. That means the rejector doesn’t just evaluate the \`\`difficulty’’ of a sample but also compares the performance difference between the local model and the server model.
>
> b. We display the data samples from SVHN that are kept locally or sent to the server in Figure 5 and Figure 6\. From visual inspection, the samples kept locally are clearer, contrast, focused, with high-resolution, while the samples that are sent to the server are blurry, unfocused, or lower-resolution. This seems to indicate that the rejector’s measure of \`\`difficulty’’ correlates to human notions of difficulty.
>
> c. We set up an additional experiment to evaluate the impact of the imbalanced dataset and distribution drifting issues for our multi-class L2H. In this experiment, we pre-train the local model with the reduced dataset that doesn’t contain any sample from one certain class (no \`\`truck’’ in CIFAR-10 and no \`\`9’’ in SVHN). Based on that, we train the rejector and edge model with a full dataset. The results in Table 5 and Table 6 in Appendix G show that almost all the samples from the missing class are identified by rejector and sent to the server model, and the overall accuracy is relatively unaffected. This indicates that multi-class L2H can be robust to certain forms of distribution changes (i.e., introducing a new class).
>
>
> **References**:
>
> \[1\]Guo, Chuan, et al. "On the calibration of modern neural networks." *International conference on machine learning*. PMLR, 2017\.
> \[2\]Cortes, Corinna, Giulia DeSalvo, and Mehryar Mohri. "Learning with rejection." *Algorithmic Learning Theory: 27th International Conference, ALT 2016, Bari, Italy, October 19-21, 2016, Proceedings 27*. Springer International Publishing, 2016\.
> \[3\]Wu, Yu, and Anand Sarwate. "Learning To Help: Training Models to Assist Legacy Devices." *arXiv preprint arXiv:2409.16253* (2024).

---

> > ### Author Response · Authors · 2024-11-21
> > **Additional experiments uploaded**
> >
> > We have added the additional experiment results to our paper. The latest version of paper is uploaded to openreview.

---

### Meta-Review · Area_Chair_yS7w · 2024-12-10

**Metareview:**

This paper was reviewed by four experts in the field and received 6, 6, 8, 6 as the final ratings. The reviewers agreed that the setting of learning to help is interesting and meaningful, the proposed method is theoretically sound, the experimental results are encouraging, and that the paper is well-written and easy to follow.

The reviewers raised concerns that the proposed method has been evaluated only on 2 datasets (CIFAR-10 and SVHN), with relatively simple architectures (LeNet and AlexNet). To address this, the authors have included experiments on the CIFAR-100 dataset with the ViT model in the revised version of the paper. The authors have also included a rigorous proof of the surrogate loss consistency in a multi-class setting. A table has also been included to show the impact of the hyper-parameters $c_{1}$ and $c_{2}$ on the system's overall performance using the CIFAR-100 dataset with the ViT model, in response to a reviewer's comment. Concerns were also raised about the cost of training, difference in convergence rate between the asynchronous and synchronous methods, and adaptability of the client classifier to changing data distributions over time, which were all addressed convincingly by the authors in the rebuttal.

The reviewers, in general, have a positive opinion about the paper and its contributions (two of the reviewers had raised their scores from 5 to 6 after reviewing the author rebuttal). Based on the reviewers' feedback, the decision is to recommend the paper for acceptance to ICLR 2025. The reviewers have provided some valuable comments, such as the privacy risks in the client-server setup, comparison against additional baseline methods and evaluating the proposed method on larger datasets such as ImageNet. The authors are encouraged to address these in the final version of their paper. We congratulate the authors on the acceptance of their paper!

**Additional Comments On Reviewer Discussion:**

Please see my comments above.

---

### Decision · Program_Chairs · 2025-01-22

Accept (Poster)